# Protein quantitative trait locus study in obesity during weight-loss identifies a leptin regulator

Jérôme Carayol [1], Christian Chabert[1], Alessandro Di Cara[2], Claudia Armenise[2], Gregory Lefebvre[1], Dominique Langin [3], Nathalie Viguerie[3], Sylviane Metairon[1], Wim H.M. Saris[4], Arne Astrup [5], Patrick Descombes [1], Armand Valsesia[1] & Jörg Hager[1]

Thousands of genetic variants have been associated with complex traits through genome-wide association studies. However, the functional variants or mechanistic consequences remain elusive. Intermediate traits such as gene expression or protein levels are good proxies of the metabolic state of an organism. Proteome analysis especially can provide new insights into the molecular mechanisms of complex traits like obesity. The role of genetic variation in determining protein level variation has not been assessed in obesity. To address this, we design a large-scale protein quantitative trait locus (pQTL) analysis based on a set of 1129 proteins from 494 obese subjects before and after a weight loss intervention. This reveals 55 BMI-associated *cis*-pQTLs and *trans*-pQTLs at baseline and 3 *trans*-pQTLs after the intervention. We provide evidence for distinct genetic mechanisms regulating BMI-associated proteins before and after weight loss. Finally, by functional analysis, we identify and validate *FAM46A* as a *trans* regulator for leptin.

[1] Nestlé Institute of Health Sciences, EPFL Innovation Park, 1015 Lausanne, Switzerland. [2] Quartz Bio, Avenue de Sécheron 15, 1202 Geneva, Switzerland. [3] INSERM UMR1048, Obesity Research Laboratory, Institute of Metabolic and Cardiovascular Diseases, University of Toulouse, 1 avenue Jean Poulhès BP 84225, 31432 Toulouse, France. [4] Department of Human Biology, NUTRIM, School of Nutrition and Translational Research in Metabolism, Maastricht University Medical Centre, PO Box 6166200 MD Maastricht, The Netherlands. [5] Department of Nutrition, Exercise and Sports, Faculty of Science, University of Copenhagen, Nørre Alle 51, DK-2200 Copenhagen N, Denmark. Correspondence and requests for materials should be addressed to J.C. (email: Jerome.Carayol@rd.nestle.com)

Thousands of genetic variants have been associated with complex traits or diseases through genome-wide association studies (GWAS)[1]. However, the mechanisms of action by which they influence traits or diseases are often unclear since most of these variants are not functional, generally located in intergenic regions or surrounding genes of unknown function. Recently, the largest GWAS meta-analysis of body mass index (BMI) performed in 339,224 individuals identified 97 BMI-associated[2] common variants. However, all of these studies have been performed at the population level apart from a few candidate gene studies conducted in obese individuals during a weight loss intervention[3,4]. Availability of high-throughput omics technologies like proteomics and transcriptomics combined with genetic variants may provide new insights into the genetic mechanisms of complex traits. A large number of expression quantitative trait (eQTL) analyses investigated the role of common genetic variants on gene expression in complex traits yielding a better understanding of their underlying mechanism[5].

Proteins as the main working blocks of metabolism are good proxies of the metabolic state of an organism. Likewise, changes of protein levels during interventions can provide insights into the level of response and sometimes predict long-term outcomes[6]. Thus, proteome analysis holds the promise to provide new insights into the understanding of mechanisms underlying diseases. This is especially true for metabolic diseases including obesity[7]. Indeed, proteomics has already provided some promising results for the understanding of molecular mechanisms and pathogenesis of obesity and related traits[6,8,9]. These studies have shown that the levels of many proteins vary significantly between obese and normal weight individuals. More importantly, many proteins have been shown to be differentially expressed in plasma of obese individuals before and after a weight loss intervention[6]. Weight loss and maintenance are hallmarks of treating obesity and in the prevention of obesity related co-morbidities like type 2 diabetes and cardiovascular disease[10,11]. However, the capacity to lose weight and keep the lost weight off is highly variable among individuals and so are their protein profiles[6].

The role of genetic variation in determining protein level variation has not been assessed in obesity. Until recently pQTL analyses were limited to modest sets of proteins in cohorts of moderate size but recent access to high-throughput technologies present the opportunity to perform large genome-wide pQTL analyses. To date only few large-scale pQTL studies have been reported and only two of them performed in mice and *Drosophila* investigated protein changes under different intervention conditions[12,13].

To investigate how genetic variation affects protein levels both at baseline and during a low-calorie-induced weight loss, we designed a pQTL analysis based on a set of 1129 proteins available from 494 obese participants from the DIOGenes (DIet, Obesity, and Genes) intervention trial[14]. We used additional information from transcriptomics available for a subset of 151 participants to perform local eQTL analyses in an effort to identify the causal mediator of distant *trans*-pQTL's. This study identifies genetic factors contributing to the variation of protein levels in a complex trait on a large-scale. The results of the integration of different omics data provide evidence for distinct genetic mechanism regulating BMI-associated proteins before and after a dietary-weight loss intervention.

## Results

**Proteome-wide association study.** At baseline, 192 proteins were significantly associated with BMI (Fig. 1a and Supplementary Data 1) with the majority ($n = 117$) showing negative correlations (i.e., low protein expression associated to high BMI). Leptin,

widely investigated in obesity[15], displayed the strongest association with BMI at baseline. Other proteins previously linked to obesity or related traits included CRP[16], IGFBP-1[17], kalistatin[18], factor H[19], or antithrombin III[20]. Proteins associated with BMI for the first time in our study included UNC5F4, a receptor for netrin involved in cell migration and angiogenesis[21]. Netrin-1 likely plays a key role in the retention of macrophages in the visceral white adipose tissue in obesity, promoting insulin resistance, and chronic inflammation[22].

Changes of BMI during the 8 week low-calorie dietary (LCD) weight loss were associated with a variation in the levels of 104 proteins (Fig. 1b and Supplementary Data 2), of which 51 showed negative correlations and 53 positive correlations with degree of weight loss respectively. Proteins associated with weight loss known to be involved in obesity or related traits included leptin[15], growth hormone receptor[23], TIG2[24] (chemerin), insulin-like growth factor-binding protein-2 (IGFBP2)[25] or sex hormone-binding globulin[26] (SHBG). Proteins we identified as potentially linked to obesity for the first time included the RET proto-oncogene (granulysin). Granulysin is present in cytotoxic granules of T lymphocytes and natural killer cells. Lynch et al.[27] observed decreased circulating natural killer and cytotoxic T lymphocytes in obese patients compared to lean controls.

Forty-two proteins were associated both with BMI at baseline and after the LCD intervention, respectively, a significant excess of shared proteins for the two time periods (hypergeometric test's $p = 1.9 \times 10^{-9}$). Leptin displayed the strongest association with BMI, both at baseline and during the intervention (Fig. 1c). Insulin-like growth factor-binding protein 1 (IGFBP-1), UNC5H4 or antithrombin III displayed stronger association with BMI at baseline than during the intervention whereas an inverse tendency was observed for IGFBP2 or SHBG. Recently, a proteome-wide study tested the association between BMI and 3,620 plasma proteins in 3,301 healthy individuals from the general population using the SOMAscan technology[28]. In this external study, summary statistics were available for 38 of our 42 BMI-associated proteins (Supplementary Table 1). All of our 38 proteins demonstrated significant association with BMI (with Bonferroni corrected $p$ value < 0.05) in this large data set. Association with BMI was in the same direction for all but two proteins, angiopoietin-2 and GDF-11.

**Functional networks for obese-specific proteins during LCD.** Correlation heatmaps for the 42 proteins associated with BMI at baseline and BMI change during LCD displayed similar patterns before and during the intervention. However, correlations were generally strongest during the intervention (Fig. 2).

Canonical pathway analysis identified the coagulation system as modulated by this subset of 42 protein-coding genes ($p = 2.53 \times 10^{-4}$ and corrected $p = 0.049$). Ingenuity pathway analysis (IPA) identified three major networks of interacting proteins. The largest network (Supplementary Fig. 1a) included 20 proteins and was predicted to be involved in tissue morphology, inflammatory response and cardiovascular diseases including proteins encoded by genes already associated with obesity (*INS*, *LEP*, *SERPINE1*, *IL1RAP*, and *RARRES2*). The second network ($n = 10$ proteins) was connected to organismal injury and abnormalities, and cellular and embryonic development (Supplementary Fig. 1b), the third network ($n = 7$ proteins) to developmental disorders, endocrine system disorders, hereditary disorders (Supplementary Fig. 1c).

**pQTL association results at baseline.** GWAS pQTL analysis of 192 BMI-associated proteins at baseline identified 57 pQTL's: 34

(Table 1) and 22 (Table 2) with expression associated with 102 *cis*-pQTL SNPs and 45 *trans*-pQTL SNPs, respectively. Adding BMI as a confounder in the analysis did not affect these results (Tables 1 and 2). The majority of SNPs were located within intronic (31%) or intergenic (50%) regions in similar proportion between *cis*-acting and *trans*-acting SNPs (Table 3). *Cis*-acting pQTLs were significantly enriched in variants in 5′ upstream ($p = 6.1 \times 10^{-4}$) and coding ($p = 2.2 \times 10^{-4}$) regions.

The strongest *cis*-pQTL association signals were located in the interleukin proteins IL-1R AcP (rs6444442, $p = 4.6 \times 10^{-66}$) and IL-16 (rs4778636, $p = 1.0 \times 10^{-35}$; Table 1) and the apolipoprotein SAA (rs7950019, $p = 4.8 \times 10^{-45}$). As expected, *trans*-pQTL association signals were generally weaker than *cis*-pQTLs (Table 2) but were still highly significant for sE-Selectin protein levels that have been linked to obesity[29]. The most significant *trans*-pQTL SNP for sE-Selectin (rs495828, $p = 9.0 \times 10^{-40}$; Supplementary Fig. 2a) was located within the *ABO* gene. This SNP was in complete linkage disequilibrium (pairwise linkage disequilibrium (LD) $r^2 = 1.00$) with rs651007, already associated with variation of sE-Selectin levels in two large Caucasian cohorts[30]. We also observed a nominally significant eQTL association within the same region of the *ABO* gene (Supplementary Fig. 2b) at baseline (eQTL association signal for rs630510, $p = 3.0 \times 10^{-4}$ in moderate LD $r^2 = 0.27$ with rs495828 pQTL SNP). Based on the GTEx database, sE-Selectin *trans*-pQTL SNP rs495828 was associated with the expression of *ABO* gene in visceral and subcutaneous adipose tissues. Among others, an interesting *trans*-pQTL is for calpastatin on chromosome 10. Calpastatin is an endogenous calpain inhibitor and is involved in inflammatory processes[31]. The strongest *trans*-pQTL for this protein lies in the carboxypeptidase N subunit 1 (*CPN1*) gene. Transcriptomics data were not available for *CPN1*. However, we detected a *cis*-eQTL association between the most significant pQTL SNP, rs11599750 (Table 2), and gene expression at baseline ($p = 9.9 \times 10^{-4}$) for the *CWF19L1* gene. This gene is located 150Kb downstream from *CPN1* (Supplementary Fig. 3).

At baseline, we identified pQTLs acting on the levels of multiple proteins. One locus on chromosome 3 was significantly associated with the levels of 3 distinct proteins (Supplementary Fig. 4a–c): PUR8 encoded by the *ADSL* gene; ferritin encoded by *FTH1* and finally CA1, a carbonic anhydrase (CA). Baseline expression levels were significantly correlated between all three proteins (Spearman correlation test with $p < 1.0 \times 10^{-6}$ and Spearman $\rho > 0.4$ for all pairwise correlations). The pQTL region included the gene for *KCNAB1*, a regulator of insulin secretion, neurotransmitter release and smooth muscle contraction, and the *TIPARP* gene, which encodes a poly (ADP-ribose) polymerase involved in DNA repair. Results from our eQTL analysis for all genes in the region identified *TIPARP* as the gene with the strongest association signal (Supplementary Fig. 4d). This gene was identified in a genome-wide meta-analysis as a potential candidate for the regulation of circulating leptin levels and its expression was increased in subcutaneous adipose tissue and liver of mice fed with a high-fat diet[32].

Two studies recently published genome-wide pQTL studies in population-based cohorts[28,33]. The study by Suhre et al.[33], analyzed the population-based KORA (Cooperative Health Research in the Region of Augsburg) for discovery and the Qatar

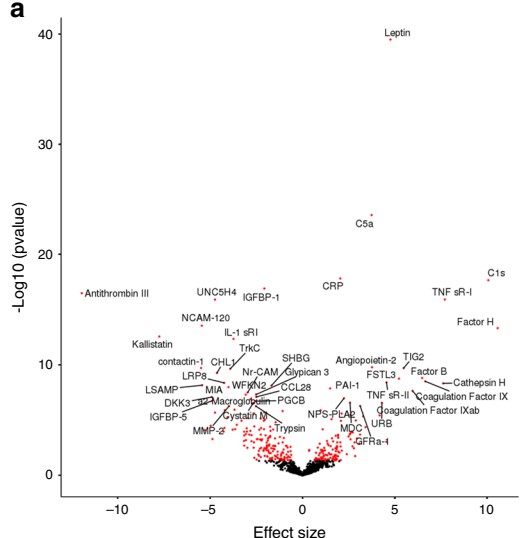

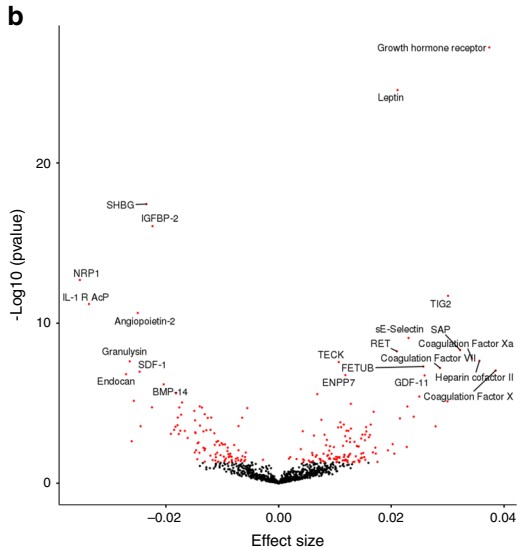

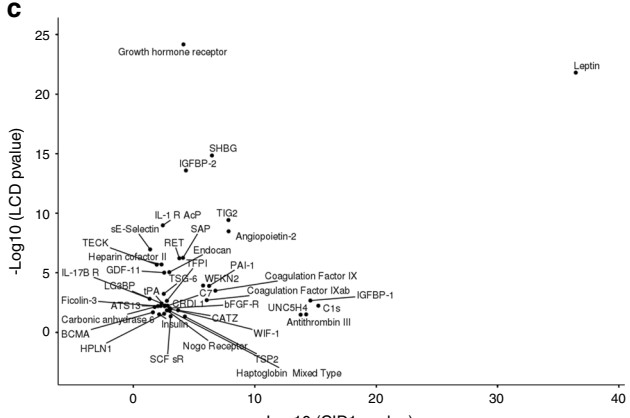

**Fig. 1** Association analysis between proteins expression and BMI. **a** Volcano plots for 192 proteins with expression associated to BMI at baseline and **b** 104 proteins for which change in expression level was associated to BMI change during LCD intervention. *x*-axis represents the effect size from the association test and *y*-axis the corresponding –log10 *p* value from a linear regression adjusted for age, gender, and center. *p* value were colored in red for proteins with significant association. A positive effect characterizes a protein expression decrease with weight loss while a negative effect depicts an increase of protein expression with weight loss during LCD intervention. Proteins with strong association to BMI change ($p$ value $< 1.0 \times 10^{-6}$) are named. **c** $p$ values from association tests for 42 proteins associated at baseline and during LCD. Each dot corresponds to a protein with –log10 $p$ value association test at baseline (*x*-axis) and during LCD (*y*-axis)

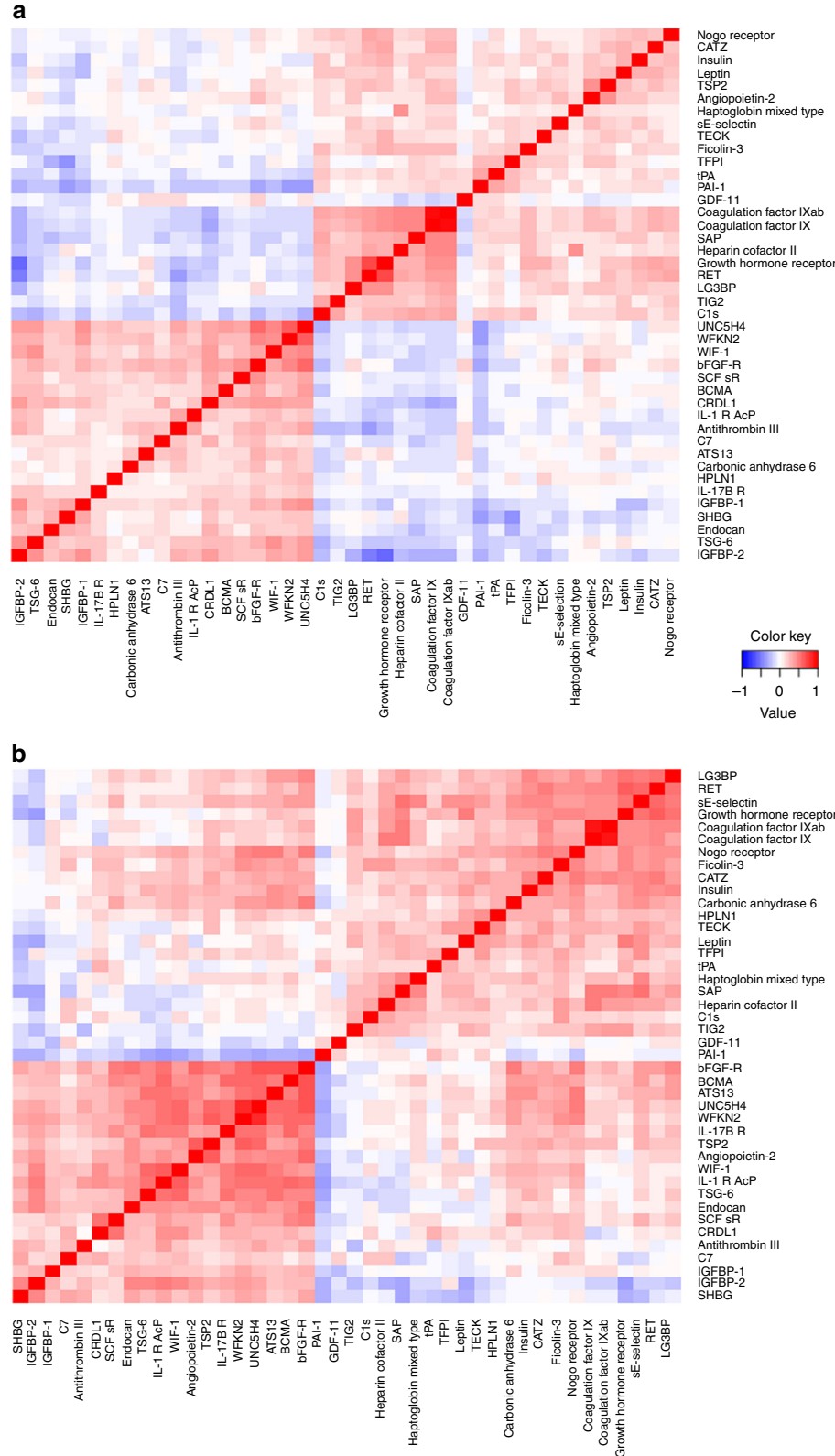

**Fig. 2** Proteins correlation heatmaps. Correlation for 42 proteins whose expression was associated to BMI at both time periods. Pairwise Spearman correlation between proteins computed using (**a**) expression level residuals at baseline and (**b**) expression fold change residuals during LCD intervention. Residuals were computed regressing proteins level at baseline or proteins expression fold change during LCD on confounding cofactors (age, gender, and center)

**Table 1 Cis-acting SNP association results for 34 proteins with significant pQTL signal at baseline**

| PROTEIN | SNP | Chr | Position | MA | MAF | Coef | SE | p value | Adjusted p value |
|---|---|---|---|---|---|---|---|---|---|
| IL-1 R AcP | rs6444444 | 3 | 190346231 | A | 0.16 | 0.66 | 0.039 | $4.60 \times 10^{-66}$ | $1.05 \times 10^{-69}$ |
| SAA | rs7950019 | 11 | 18278912 | G | 0.37 | 1.53 | 0.109 | $4.81 \times 10^{-45}$ | $2.52 \times 10^{-44}$ |
| IL-16 | rs4778636 | 15 | 81591639 | A | 0.07 | −0.48 | 0.039 | $1.00 \times 10^{-35}$ | $7.68 \times 10^{-37}$ |
| MIA | rs2607426 | 19 | 41274713 | G | 0.06 | 0.49 | 0.04 | $3.95 \times 10^{-34}$ | $8.55 \times 10^{-36}$ |
| HCC-4 | rs11080369 | 17 | 34305164 | C | 0.05 | −0.79 | 0.069 | $1.58 \times 10^{-30}$ | $6.97 \times 10^{-31}$ |
| kallikrein 12 | rs3745540 | 19 | 51535130 | A | 0.38 | 0.33 | 0.029 | $8.36 \times 10^{-29}$ | $4.54 \times 10^{-28}$ |
| sICAM-5 | rs281440 | 19 | 10400304 | G | 0.24 | −0.3 | 0.028 | $6.75 \times 10^{-27}$ | $1.95 \times 10^{-27}$ |
| NPS-PLA2 | rs10732279 | 1 | 20305065 | C | 0.27 | 0.38 | 0.036 | $3.44 \times 10^{-26}$ | $5.76 \times 10^{-27}$ |
| ATS13 | rs4962144 | 9 | 136275141 | A | 0.1 | −0.34 | 0.035 | $7.08 \times 10^{-23}$ | $1.84 \times 10^{-22}$ |
| ECM1 | rs13294 | 1 | 150484987 | A | 0.44 | −0.2 | 0.02 | $6.65 \times 10^{-22}$ | $4.38 \times 10^{-22}$ |
| PARC | rs972317 | 17 | 34371659 | G | 0.12 | 0.5 | 0.052 | $6.90 \times 10^{-22}$ | $4.61 \times 10^{-23}$ |
| Carbonic anhydrase 6 | rs3765964 | 1 | 9034421 | A | 0.39 | 0.44 | 0.052 | $3.37 \times 10^{-17}$ | $3.20 \times 10^{-17}$ |
| Factor H | rs1329428 | 1 | 196702810 | T | 0.43 | 0.08 | 0.009 | $4.94 \times 10^{-16}$ | $1.22 \times 10^{-16}$ |
| TECK | rs11671930 | 19 | 8117308 | C | 0.16 | −0.5 | 0.063 | $1.38 \times 10^{-15}$ | $1.23 \times 10^{-15}$ |
| Haptoglobin mixed type | rs2000999 | 16 | 72108093 | A | 0.2 | −0.49 | 0.062 | $3.23 \times 10^{-15}$ | $2.56 \times 10^{-16}$ |
| FCN2 | rs7851696 | 9 | 137779091 | T | 0.11 | −0.38 | 0.049 | $4.61 \times 10^{-15}$ | $1.07 \times 10^{-15}$ |
| Kallistatin | rs5510 | 14 | 95033356 | T | 0.24 | 0.11 | 0.014 | $8.78 \times 10^{-14}$ | $2.27 \times 10^{-15}$ |
| Cripto | rs3806702 | 3 | 46617502 | C | 0.27 | 0.14 | 0.019 | $9.53 \times 10^{-14}$ | $2.57 \times 10^{-13}$ |
| CYTN | rs6036507 | 20 | 23721992 | A | 0.3 | 0.31 | 0.042 | $2.25 \times 10^{-13}$ | $2.04 \times 10^{-13}$ |
| WFKN2 | rs4586493 | 17 | 48924465 | A | 0.28 | 0.18 | 0.026 | $7.84 \times 10^{-12}$ | $3.25 \times 10^{-12}$ |
| CYTT | rs6036507 | 20 | 23721992 | A | 0.3 | 0.3 | 0.045 | $1.02 \times 10^{-11}$ | $2.04 \times 10^{-13}$ |
| Contactin-5 | rs1461674 | 11 | 99180043 | T | 0.08 | 0.23 | 0.035 | $1.18 \times 10^{-11}$ | $3.21 \times 10^{-12}$ |
| MIP-1a | rs972317 | 17 | 34371659 | G | 0.12 | 0.35 | 0.051 | $1.33 \times 10^{-11}$ | $4.61 \times 10^{-23}$ |
| Factor B | rs541862 | 6 | 31916951 | C | 0.08 | 0.15 | 0.023 | $3.39 \times 10^{-11}$ | $6.73 \times 10^{-12}$ |
| Proteinase-3 | rs2074639 | 19 | 840984 | T | 0.22 | 0.22 | 0.033 | $5.74 \times 10^{-11}$ | $1.39 \times 10^{-10}$ |
| TIG2 | rs10282458 | 7 | 150045302 | A | 0.25 | 0.12 | 0.018 | $2.11 \times 10^{-10}$ | $1.34 \times 10^{-9}$ |
| LD78-beta | rs972317 | 17 | 34371659 | G | 0.12 | 0.2 | 0.032 | $3.27 \times 10^{-10}$ | $4.61 \times 10^{-23}$ |
| TSG-6 | rs6715902 | 2 | 152201469 | A | 0.34 | 0.18 | 0.031 | $3.23 \times 10^{-9}$ | $8.92 \times 10^{-9}$ |
| LBP | rs2284878 | 20 | 36785127 | A | 0.06 | −0.36 | 0.062 | $4.07 \times 10^{-9}$ | $5.92 \times 10^{-9}$ |
| C7 | rs3805221 | 5 | 40965318 | T | 0.24 | −0.12 | 0.02 | $6.73 \times 10^{-9}$ | $7.45 \times 10^{-9}$ |
| Factor I | rs7439493 | 4 | 110656730 | A | 0.42 | 0.06 | 0.01 | $1.46 \times 10^{-8}$ | $4.69 \times 10^{-9}$ |
| Kallikrein 7 | rs1654523 | 19 | 51485194 | C | 0.08 | −0.24 | 0.044 | $5.57 \times 10^{-8}$ | $1.75 \times 10^{-8}$ |
| TSP2 | rs3763267 | 6 | 169571693 | C | 0.09 | 0.29 | 0.053 | $5.66 \times 10^{-8}$ | $5.20 \times 10^{-8}$ |
| CHL1 | rs6769747 | 3 | 244949 | G | 0.1 | 0.16 | 0.03 | $1.37 \times 10^{-7}$ | $6.81 \times 10^{-6}$ |

pQTL association results are provided with (adjusted p value) and without (p value) adjusting for BMI
SNP single-nucleotide polymorphism, Chr chromosome, MA minor allele, MAF minor allele frequency, Coef estimated association coefficient, SE standard error

Metabolomics Study on Diabetes (QMDiab) for replication. Eight of our pQTL SNPs were significant in KORA and four of them successfully replicated in QMDiab. We also compared our results with the study by Sun et al.[28]. This highlighted two additional pQTLs (Supplementary Data 3).

Since these comparisons were limited to the presence of the same SNP marker, we extended it to a locus-based comparison (i.e. QTL signals). This was done by extending association signals to pQTL SNPs and SNPs in moderate LD ($r^2 > 0.2$ based on 1000 Genomes Project genotype European reference population). This approach showed that 39 and 34 of our QTL signals were in common with those reported by Suhre et al.[33] or Sun et al.[28] respectively. Specifically, 22 QTL signals were shared between all three studies (Supplementary Data 4). pQTL signals, we replicated included IL-1R AcP described above and CA6, discussed later in this paper.

**pQTL associations under a low-calorie diet challenge.** Changes in 104 proteins were associated with BMI variation during the LCD intervention. Our pQTL analysis identified three proteins significantly regulated by four SNPs. All were trans-acting pQTLs (Table 4) and intergenic except rs1544241, located in the 5′-upstream region of the pseudogene *RP11-672L10.5*. Inclusion of BMI as a confounder in the statistical model did not impact association results (Table 4).

One SNP located in a large intergenic region on chromosome 6 displayed the strongest pQTL association signal. This SNP, rs13200531, was associated with angiotensinogen protein level changes ($p = 3.88 \times 10^{-10}$). Expressed in adipose tissue, angiotensinogen contributes to diet-induced obesity[34]. In our DIOGenes eQTL study, the SNP did not display association signals with any gene in the region. SNP rs1544241 was identified as a trans-acting pQTL for caspase-10 a protein involved in apoptosis reported in adipocytes of obese subjects[35]. eQTL association results from genes surrounding the locus did not highlight any potential regulatory gene. Two SNPs were associated with changes in leptin levels during the LCD as detailed below. During LCD intervention, leptin was correlated with caspase-10 (Spearman $\rho = 0.12$ and $p = 9 \times 10^{-3}$) and angiotensinogen (Spearman $\rho = 0.23$ and $p = 1.55 \times 10^{-7}$). Leptin is known to inhibit apoptosis via downregulation of caspase-10[36] and was found significantly correlated with plasma angiotensinogen in healthy normotensive subjects[37].

**pQTL effects on CA6, chemerin, and leptin protein expression.** Of the 192 proteins associated with BMI at baseline, 42 also showed significant association during LCD. In total, 14 (Supplementary Data 5) of these 42 displayed significant pQTL signals at baseline but not after the LCD. Nonetheless, we considered that two of them, CA6 (gustin) and chemerin, deserved further investigation because of the biological relevance to

**Table 2 *Trans*-acting SNP association results for 22 proteins with significant pQTL signal at baseline**

| Protein | SNP | Chr | Position | MA | MAF | Coef | SE | *p* value | Adjusted *p* value |
|---|---|---|---|---|---|---|---|---|---|
| sE-Selectin | rs495828 | 9 | 136154867 | T | 0.2 | −0.5 | 0.04 | $8.98 \times 10^{-40}$ | $1.13 \times 10^{-39}$ |
| Calpastatin | rs11599750 | 10 | 101805442 | T | 0.4 | −0.2 | 0.02 | $3.78 \times 10^{-12}$ | $2.49 \times 10^{-12}$ |
| FCN2 | rs7103402 | 11 | 114391633 | T | 0.1 | −0.3 | 0.05 | $7.27 \times 10^{-11}$ | $1.69 \times 10^{-10}$ |
| MP2K2 | rs12876365 | 13 | 113911651 | T | 0.2 | −0.2 | 0.03 | $3.37 \times 10^{-10}$ | $1.27 \times 10^{-10}$ |
| HAI-1 | rs1886748 | 1 | 39333775 | A | 0.4 | −0.2 | 0.03 | $6.97 \times 10^{-10}$ | $5.62 \times 10^{-10}$ |
| IDS | rs7661253 | 4 | 183039092 | C | 0.4 | −0.2 | 0.03 | $1.50 \times 10^{-9}$ | $3.68 \times 10^{-9}$ |
| CATZ | rs10860794 | 12 | 102217720 | A | 0.3 | −0.1 | 0.02 | $1.78 \times 10^{-9}$ | $7.39 \times 10^{-9}$ |
| 40 S ribosomal protein SA | rs7299019 | 12 | 105271085 | T | 0.1 | 0.27 | 0.05 | $1.56 \times 10^{-8}$ | $7.98 \times 10^{-9}$ |
| PUR8 | rs10513494 | 3 | 156291263 | G | 0.1 | 0.42 | 0.08 | $2.74 \times 10^{-8}$ | $6.52 \times 10^{-8}$ |
| a1-Antichymotrypsin | rs10509957 | 10 | 114053983 | A | 0.4 | −0.1 | 0.01 | $3.80 \times 10^{-8}$ | $2.46 \times 10^{-7}$ |
| Carbonic anhydrase I | rs10513494 | 3 | 156291263 | G | 0.1 | 0.42 | 0.08 | $3.93 \times 10^{-8}$ | $1.13 \times 10^{-7}$ |
| TGF-b3 | rs2058950 | 7 | 155182326 | A | 0.2 | 0.1 | 0.02 | $7.13 \times 10^{-8}$ | $1.05 \times 10^{-7}$ |
| IL-7 | rs7836269 | 8 | 13748948 | G | 0.1 | 0.31 | 0.06 | $7.66 \times 10^{-8}$ | $1.90 \times 10^{-7}$ |
| Cytochrome c | rs9325542 | 10 | 114977369 | G | 0.2 | 0.15 | 0.03 | $7.86 \times 10^{-8}$ | $2.45 \times 10^{-7}$ |
| ACTH | rs2268992 | 6 | 88183647 | G | 0.1 | 0.35 | 0.07 | $9.18 \times 10^{-8}$ | $2.27 \times 10^{-8}$ |
| MMP-16 | rs11230085 | 11 | 59578322 | A | 0.1 | 0.23 | 0.04 | $9.68 \times 10^{-8}$ | $1.25 \times 10^{-7}$ |
| Nectin-like protein 1 | rs12420752 | 11 | 21866461 | T | 0.1 | 0.31 | 0.06 | $1.18 \times 10^{-7}$ | $2.27 \times 10^{-7}$ |
| BGN | rs7309378 | 12 | 12828528 | A | 0.1 | −0.2 | 0.04 | $1.24 \times 10^{-7}$ | $7.15 \times 10^{-7}$ |
| Coactosin-like protein | rs10908893 | 9 | 92117102 | T | 0 | 0.32 | 0.06 | $1.27 \times 10^{-7}$ | $2.53 \times 10^{-8}$ |
| NKp44 | rs1432180 | 2 | 82079857 | A | 0.2 | 0.11 | 0.02 | $1.51 \times 10^{-7}$ | $2.67 \times 10^{-7}$ |
| MMP-2 | rs7935013 | 11 | 467339 | G | 0.3 | 0.09 | 0.02 | $1.52 \times 10^{-7}$ | $8.29 \times 10^{-7}$ |
| Ferritin | rs12485954 | 3 | 156336114 | C | 0.1 | 0.47 | 0.09 | $1.54 \times 10^{-7}$ | $3.40 \times 10^{-7}$ |

pQTL association results are provided with (adjusted *p* value) and without (*p* value) adjusting for BMI

*SNP* single-nucleotide polymorphism, *Chr* chromosome, *MA* minor allele, *MAF* minor allele frequency, *Coef* estimated association coefficient, *SE* standard error

obesity and strong, consistent genetic association signals during the LCD.

The first protein, CA6 (gustin), was significantly regulated by proximal common variants at baseline (rs3765964, $p = 3.37 \times 10^{-17}$; Supplementary Fig. 5). Gustin controls the salivary trophic factor and a disruption in the protein decreases taste function. Gustin is a zinc-dependent enzyme. Genetic variants in gustin are associated with individual ability to taste 6-*n*-propylthiouracil (PROP) which in turn is inversely related to BMI and salivary ionic zinc concentrations[38]. The PROP phenotype has been associated with variation in the perception and preference for fat[39], energy intake, and body weight[40,41]. The strongest pQTL signal for the protein during LCD (Supplementary Fig. 6) was in *trans* and located in the promoter region of *OR2W5*, an olfactory receptor in the nose. This SNP (rs7512601; Supplementary Fig. 7) displayed strong association with gustin protein levels ($p = 3.8 \times 10^{-7}$; Supplementary Fig. 8) but did not reach significance after multiple testing correction ($p_{corrected} = 0.10$). None of the pQTL SNPs showed an eQTLs signal for other genes in a 5 Mb surrounding region in our data.

The second protein, chemerin, was significantly regulated at baseline by *cis*-acting SNPs with the strongest signal for rs10282458 ($p = 2.11 \times 10^{-10}$, Supplementary Fig. 5). Changes in protein levels during LCD displayed its strongest association with rs11113832 ($p_{nominal} = 1.5 \times 10^{-6}$; $p_{corrected} = 0.37$; Supplementary Fig. 9), a *trans*-pQTL SNP located in the promoter region of the chemerin receptor gene *CMKLR1*. Adding information from imputed SNPs, the region shows enrichment of an association signal (Supplementary Fig. 10). Our transcriptomics data did not identify cis-eQTLs for *CMKLR1* gene expression. Chemerin plays an important role in obesity as an adipokine involved in the regulation of adipogenesis and adipocyte metabolism[42].

Among the 42 proteins associated with BMI both at baseline and during the LCD, only leptin displayed a significant pQTL signal during LCD. Two tightly linked SNPs in an intergenic region on chromosome 6 were associated with leptin protein

expression changes (Fig. 3 and Supplementary Fig. 7). Surrounding genes included *BCKDHB*, involved in the catabolism of branched chain amino acids (BCAAs); and *FAM46A*, a SMAD signaling pathway related protein involved in TGF-β signaling pathway[43]. Using adipocyte expression data from our study we performed an eQTL analysis. Only the *FAM46A* gene expression displayed association with SNPs in the pQTL region. No other gene within 5 Mb around the pQTL locus displayed eQTL association signals higher than the *FAM46A* signal in our data. The biological role of *FAM46A* is largely unknown. The protein interacts with the BAG6 protein[44] involved in gene regulation, apoptosis, regulation of protein synthesis, and degradation. It has recently been shown that BAG6 splicing in subcutaneous fat is highly determined by BMI[45]. BAG6 is also involved in stress response. There is a reciprocal relationship between heat shock protein HSP70 and BAG6 suggesting that BAG6 could be a central regulator of the cellular content of HSP70[46] which is in turn positively associated with leptin in type 2 diabetes[47] and in our data ($p = 3.4 \times 10^{-4}$ from a multivariate regression adjusted for age, gender, center, and BMI change during LCD intervention). pQTL association signals in the regulatory region of *FAM46A* were also observed for HSP70 (Supplementary Fig. 11a). Adjusting for leptin expression change as a confounding cofactor did not affect pQTL results (Supplementary Fig. 11b). In other words, the HSP70 pQTL signal in the *FAM46A* regulatory region was leptin-level independent.

The reciprocal relationship between *FAM46A* and leptin is also apparent during adipocyte maturation. In preadipocytes *FAM46A* expression is high and decreases during maturation to a minimum at day seven. This decrease coincides with the appearance of leptin (Supplementary Fig. 12). To complete the genetic characterization of the QTL results we evaluated the directionality of pQTL and eQTL SNPs. We first focused on the 2 pQTL SNPs significantly associated with leptin change during LCD. SNP rs9344031 was positively associated with leptin protein level change ($p = 1.48 \times 10^{-7}$) and negatively associated with *FAM46A* gene expression change ($p = 4.9 \times 10^{-3}$). The same trend was found for the second pQTL SNP rs481777 although gene

**Table 3 Functional annotation of 102 *cis* and 45 *trans*-pQTL associated to proteins level at baseline**

| Function | Cis regulation | | | Trans regulation | | |
|---|---|---|---|---|---|---|
| | Count | Proportion | Corrected *p* value | Count | Proportion | Corrected *p* value |
| 3′downstream | 0 | 0 | 1 | 1 | 0.022 | 1 |
| 5′upstream | 9 | 0.088 | 0.00061 | 4 | 0.089 | 0.057 |
| 5′utr | 1 | 0.01 | 1 | 1 | 0.022 | 0.5 |
| coding | 10 | 0.098 | 0.00022 | 1 | 0.022 | 1 |
| intergenic | 30 | 0.294 | 1 | 16 | 0.356 | 1 |
| intronic | 51 | 0.5 | 1 | 22 | 0.489 | 1 |
| non-coding intronic | 1 | 0.01 | 0.22 | 0 | 0 | 1 |

*p* value from two-tailed Fisher's exact test for enrichment analysis of pQTL SNPs among regulatory elements were adjusted for multiple testing using Bonferroni correction

expression did not reach nominal significance ($p = 0.12$) likely because of the difference in sample size between the eQTL and pQTL study. In an enlarged analysis, pQTL signals were extracted in the promoter region of *FAM46A* located between 81.3 and 81.8 Mb on chromosome 6 (Fig. 3) for all genotyped and imputed SNPs. Of 1,833 SNPs in this region, 94 were identified as pQTLs and eQTLs for leptin and *FAM46A*, respectively, assuming an uncorrected 5% nominal significance level. All QTL SNPs displayed opposite association direction between leptin protein and *FAM46A* gene expression, consistent with a regulatory role of *FAM46A* on leptin.

To evaluate if *FAM46A* levels directly influenced leptin secretion, we performed knockdown and overexpression studies in the SGBS human adipocyte cell line. For knockdown experiments, a specific siRNA for the *FAM46A* gene was designed. Transfection with siRNA resulted in a 70% and 74% reduction of *FAM46A* gene expression in the basal and insulin stimulated state, respectively. FAM46A protein was reduced by 31% and 73% in the basal and insulin-induced state, respectively (Fig. 4a, b and Supplementary Fig. 13). *FAM46A* knockdown had no influence on leptin gene expression (Fig. 4c) and did not show any effect on markers of adipocyte differentiation *PPARγ* and *CEBPα* (Supplementary Fig. 14). Conversely, *FAM46A* knockdown resulted in a highly significant 49% increase of insulin stimulated leptin secretion from the adipocytes ($p = 0.0002$ one-sided *t* test for leptin increase). Under conditions without prior insulin stimulation the knockdown still showed a marginal 19% increase in leptin secretion ($p = 0.065$ one-sided test for leptin increase; Fig. 5a). To evaluate whether the effects of the knockdown of *FAM46A* were specific for leptin we tested the secretion of adiponectin. Secretion of adiponectin was not affected by the knockdown (Supplementary Fig. 15).

To overexpress *FAM46A*, SGBS cells were transfected with 0.5 μg of full length *FAM46A* cDNA. This resulted in a 21-fold increase of *FAM46A* expression (Supplementary Fig. 16). *FAM46A* overexpression resulted in a significant 28% and 24% ($p = 0.034$ and 0.038, one-sided *t* test, for both basal and insulin stimulated states, respectively) reduction of insulin and non-insulin stimulated leptin release, respectively (Fig. 5b). Overexpression did not influence *PPARγ* or *CEBPα* gene expression (Supplementary Fig. 17). Adiponectin release was not affected by *FAM46A* overexpression, both in the basal as well as the insulin stimulated state (Supplementary Fig. 18).

**Comparison of pQTLs with known GWAS loci.** Locke et al.[2] performed the largest GWAS of BMI in 339,224 individuals. Among all pQTL SNPs identified in our study, 16 were nominally ($p < 5\%$) associated with BMI in the published GWAS[2] (Supplementary Data 6). Overall, the identified pQTL SNPs were significantly enriched in SNPs nominally associated with BMI

in Locke et al.[2] (permutation test $p = 0.009$). No LCD pQTL displayed association with BMI in the published GWAS[2]. Extending the comparison to variants in LD with pQTL SNPs did not extract more BMI association results.

Inversely, pQTL signals from the 97 BMI-associated SNPs identified in Locke et al.[2] were extracted from the current study. Information from 62 SNPs was available. Missing SNPs were replaced by SNPs in LD ($r^2 > 0.8$), leading to a total of 83 SNPs available for analysis. For each protein, pQTL association for these SNPs was calculated and *p* values corrected for multiple testing using the Benjamin–Hochberg method[48].

Thirty-one out of 83 GWAS SNPs displayed nominal association with expression levels of one protein at least at baseline (Supplementary Data 7). All were identified as *trans*-regulating pQTLs including pQTLs for *FTO* and *CRP*. During LCD, 16 SNPs were identified as potential pQTLs (Supplementary Data 8). However, none of the SNPs reached significance at the proteome-wide level at baseline or during LCD.

## Discussion

Large-scale pQTL studies looking at different metabolic states are rare. In the current study we addressed the question of how genetic variation contributes to protein level differences in obesity by performing proteome quantitative trait analyses and comprehensive genotyping in a large cohort of obese individuals before and after a weight loss intervention. We showed that one third of 192 proteins associated with BMI at baseline are regulated by common genetic variants. To our knowledge, this is the first pQTL analysis in obese subjects, following a dietary intervention. Previous studies focused on association between baseline protein levels and BMI in the general population. About a third of the pQTL signals were in *trans*, meaning that these pQTLs harbor potential regulators of the associated proteins. This number of distant pQTLs correlates well with recently published pQTL studies in humans[28,33] and in mice[13]. The pQTL analysis performed in mouse strains also confirmed that distant (*trans-*) pQTLs can identify causal intermediates or distant proteins regulating levels of the pQTL associated protein[13].

Some of our baseline *cis*-pQTLs are for proteins already implicated in obesity previously. For example, IL-1RAP levels have been reported as being lower in obese compared to normal weight individuals and a genetic association with protein levels was reported in a candidate gene study[49]. We identified a new pQTL for IL-16 levels. IL-16 is a chemoattractant and modulator of lymphocyte activation. Infiltration of adipose tissue by lymphocytes and the resulting local inflammation are thought to be involved in promoting insulin resistance. Indeed, IL-16 levels in plasma were positively correlated

**Table 4 _Trans_-acting SNP association results for three proteins with significant pQTL signal during LCD**

| Protein | SNP | Chr | Position | MA | MAF | Coef | SE | _p_ value | Adjusted _p_ value |
|---|---|---|---|---|---|---|---|---|---|
| Angiotensinogen | rs13200531 | 6 | 23242826 | G | 0.07 | -0.22 | 0.036 | $3.88 \times 10^{-10}$ | $1.19 \times 10^{-10}$ |
| Leptin | rs9344031 | 6 | 81400749 | G | 0.10 | 0.31 | 0.059 | $1.48 \times 10^{-7}$ | $1.11 \times 10^{-7}$ |
| Leptin | rs481777 | 6 | 81578293 | G | 0.24 | 0.22 | 0.042 | $1.38 \times 10^{-7}$ | $5.57 \times 10^{-6}$ |
| Caspase-10 | rs1544241 | 18 | 855264 | T | 0.44 | 0.09 | 0.017 | $1.57 \times 10^{-7}$ | $2.36 \times 10^{-7}$ |

MAF, pQTL _p_ value (pvalue) and corresponding tested protein are provided
_SNP_ single-nucleotide polymorphism, _Chr_ chromosome, _MA_ minor allele, _MAF_ minor allele frequency, _Coef_ estimated association coefficient, _SE_ standard error

with CRP, a marker for inflammatory status (Spearman $\rho = 0.17$, $p = 1.1 \times 10^{-4}$).

Our strongest _trans_-pQTL signal at baseline was for sE-Selectin, previously shown to be associated with obesity. The pQTL SNPs were located within the ABO gene on chromosome 9 which also displayed an eQTL signal using DIOGenes transcriptomics information. sE-Selectin plays a role in cell adhesion in inflammatory processes. Our findings confirm that genetically determined variation of ABO gene expression alter the abundance of sE-Selectin[30], probably mediated through changing glycosylation of the protein.

Among other strong trans-pQTL signals we would like to highlight the trans-pQTL for calpastatin on chromosome 10. Calpastatin is an endogenous calpain inhibitor, involved in inflammatory processes[31]. This protein has been reported as a potential activator of adipose-derived stromal stem cells that have been associated with the occurrence of certain cancers[50]. The same study also reported that obesity leads to a dysregulation of calpastatin. The strongest trans-pQTL SNP for this protein lies in the carboxypeptidase N subunit 1 (_CPN1_) gene, a metalloprotease that regulates other proteins through specific cleavage of amino acids from the C terminal. CPN1 has been shown to be a pleiotropic regulator of inflammation[51]. Expression for CPN1 was not available in our data. However, a strong association signal was identified in our eQTL study for the more distant gene _CWF19L1_, predicted to be involved in cell cycle control.

We also identified _trans_-pQTLs that influence the levels of several proteins. A locus on chromosome 3 was significantly associated with the levels of PUR8 encoded by the ADSL gene, involved in uric acid metabolism; ferritin encoded by FTH1, involved in iron storage and CA1, a carbonic anhydrase. Recently, Gao et al.[52] demonstrated that leptin transcription was regulated by iron, making serum ferritin levels one of the best predictors of leptin serum levels under physiological conditions. Comparison of results from our eQTL analysis for all genes in the pQTL region identified _TIPARP_ as the gene with the strongest association signal. This gene was already identified in a genome-wide meta-analysis as a candidate for the regulation of circulating leptin levels and its expression was increased in subcutaneous adipose tissue and liver of mice fed with high-fat diet[32].

The functional relationship between these proteins is not known. However, both uric acid and ferritin (the levels of which are determined in part by the associated proteins) have been shown to be markers for non-alcoholic fatty liver disease (NAFLD) and its progression, a major and frequent complication in obese individuals[53]. CA is also a major regulator of uric acid levels and CA inhibitors are frequently used as diuretics and in the treatment of gout. It is intriguing that expression levels of the _TIPRAP_ gene are elevated in the livers of high-fat fed mice. Thus, one may speculate that the pQTL variant could be involved in the etiology of NAFLD through the regulation of the levels of these three proteins.

One of the most interesting and novel findings of our study is that an intervention like a low-calorie weight loss, identified pQTLs that are not detectable at baseline. All of the pQTLs detected under the intervention were in _trans_, providing evidence that a shift in homeostasis affected protein intermediates that may be activated by the intervention and regulate the pQTL associated protein. The example we wish to highlight is the _trans_-pQTL for leptin, one of the most studied proteins in obesity. Leptin levels are associated with BMI and change during weight loss[15]. However, surprisingly little is known about the regulation of leptin secretion. The pQTL on chromosome 6 is located between two genes, _BCKDHB_, which codes for a protein implicated in BCAAs catabolism, and _FAM46A_, a protein belonging to the superfamily of nucleotidyltransferase fold proteins. Both proteins are expressed in adipocytes. Analysis of the pQTL SNPs showed a consistent _cis_-eQTL signal for _FAM46A_ using transcriptomics data from 151 DIOGenes subjects but not for _BCKDHB_ or any other gene within a 5 Mbase interval around the pQTL. Thus _FAM46A_ seemed the most likely intermediate for the leptin pQTL. The exact function of _FAM46A_ is unknown. It has been associated with the SMAD pathway and TGF-beta signaling and also with heat shock protein (Hsp70)-mediated stress response. _FAM46A_ was upregulated during the low-calorie intervention, which causes a strong metabolic stress response. None of the two leptin associated pQTL SNPs or any SNPs in the surrounding region was found associated with leptin in the Kilpelainen et al.[32] meta-analysis. This study used population-based cohorts and tested association between common genetic variants and circulating leptin levels at baseline whereas our pQTL associated with leptin change only during the weight loss intervention. This may be due to a specific induction of the FAM46A gene or protein during weight loss. The reciprocal relationship between _FAM46A_ and leptin is also apparent in the shift of gene expression during adipocyte differentiation. Thus, the external stimulus and drastic change would be necessary to identify the relationship. We believe that this is a first example that an environmental cue (like a low-calorie diet) that will induce large changes in gene and protein expression can lead to the identification of pQTL signals not detectable at baseline homeostasis.

To determine whether _FAM46A_ variation directly influenced leptin levels, we used a siRNA approach to knockdown the _FAM46A_ gene in the human adipocyte cell line SGBS. We show that the reduction of _FAM46A_ expression results in higher leptin protein levels secreted from SGBS cells both at baseline and after insulin stimulation compared with the wildtype, although the baseline changes do not reach nominal significance, most likely because of the low leptin secretion levels under non-stimulated conditions. Leptin gene expression remained unaltered. This indicates that _FAM46A_ has indeed a negative regulatory effect on leptin and that this regulation happens at the protein level. It would also indicate that the SNPs affecting _FAM46A_ gene expression might at least in part be responsible for the variability in leptin response seen during the LCD

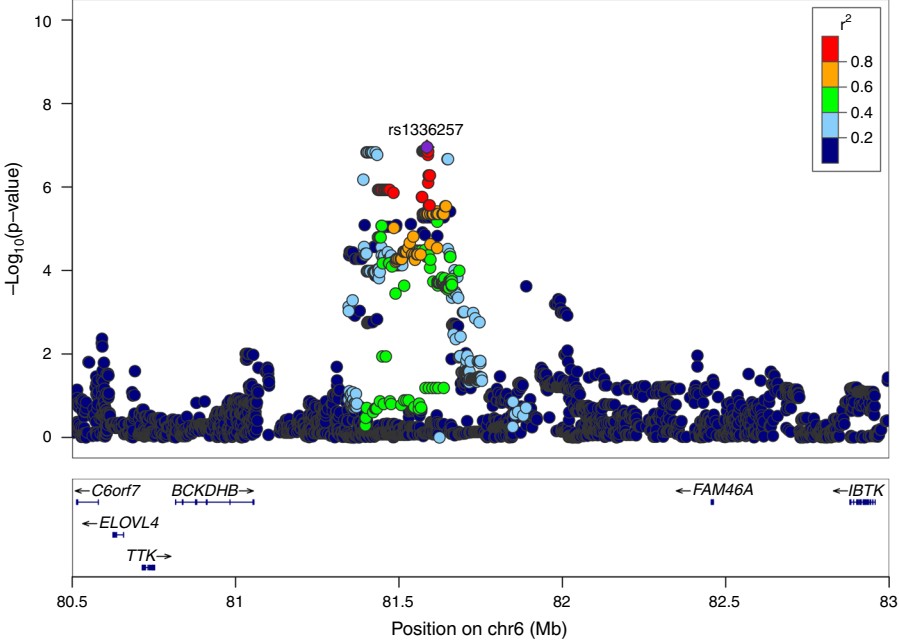

**Fig. 3** Locus-specific plot for leptin associated pQTL SNPs located in between *FAM46A* and *BCKDHB* genes. Association plot produced by LocusZoom for distal genotyped and imputed SNPs associated to leptin protein expression change during LCD. SNPs *p* values are plotted after –log10 transformation with scale on the *y*-axis and colors reflect pairwise linkage disequilibrium with the most associated SNP in the region (purple dot) based on the 1000 genomes EUR data set

intervention. Conversely, overexpression of *FAM46A* resulted in a significant reduction of leptin secretion in line with our hypothesis. Both knockdown as well as overexpression did not alter *PPARγ* and *CEBPα* levels or adiponectin secretion, indicating that the changes seen in leptin secretion are not due to effects of *FAM46A* on adipocyte differentiation and that the effects seem to be specific for leptin.

Of 42 proteins associated both with BMI at baseline and during LCD, 38 were already investigated in a population-based study and identified as correlated to BMI[28]. Comparison of pQTL signals with two large independent studies[28,33], conducted in the general population, further supported our pQTL findings and highlighted 22 pQTL signals that were common between all three studies. Both our functional assays and comparison of association signals with studies of slightly different design, lend support to our results and stress the importance of our identified pQTL signals and their potential role in regulation of proteins like *FAM46A* and gustin.

Our study has some limitations. Proteomics information was available from plasma while eQTL analyses were carried out in samples from adipose tissue biopsies in a smaller subset of participants of the DIOGenes study. This limitation of tissue availability between omics data did not allow us to perform tissue specific pQTL-eQTL comparative analyses like in Chick et al[13]. We therefore used a two-step pQTL-mapping strategy combined with eQTL mapping to limit the multiple testing burden and gain greater insight in the mechanism underlying the GWAS signal. This approach enabled us to confirm local transcriptional effects on *cis*-pQTL's in some cases and also to identify the most likely regulatory transcripts for *trans*-pQTLs like in the case of *FAM46A*. Similarly to eQTL[5], effect sizes of *cis*-acting pQTLs are expected to be larger than those of *trans*-acting pQTLs leading to a higher number of *cis*-pQTLs as observed at baseline in the current and previous studies[13]. Thus, the complete absence of *cis*-acting SNP's during the LCD intervention was an unexpected result. Extending our pQTL analysis to all proteins including those not correlated to BMI during LCD identified 42 proteins.

All of them were regulated by trans-acting genetic variants. The same observation held true when relaxing the cutoff for corrected association *p* values to 0.20 with 175 proteins identified associated with *trans*-acting pQTL SNPs only ($n = 233$ SNPs with uncorrected *p* values ranging from $1.6 \times 10^{-10}$ to $7.7 \times 10^{-7}$). We believe this to be an important finding as it indicates that an intervention, like the low-calorie diet in our study, may be especially suited to identify causal intermediates of such distant pQTL signals that are not detectable at baseline because these intermediates are only activated by the intervention.

In conclusion, we performed the first large-scale pQTL study in a complex disease reported to date. Using data from a dietary intervention study in obese patients we identified multiple *cis*-pQTL and *trans*-pQTL signals associated with BMI at baseline and after the intervention. Finally we provide evidence for *FAM46A* being a negative regulator of leptin in human adipose tissue.

## Methods

**Dietary intervention study**. This study is part of the Diet, Obesity, and Genes (DIOGenes) intervention study. A complete description of this dietary intervention is provided in Larsen et al.[14]. Briefly, 932 Caucasian overweight or obese subjects were recruited in clinical centers from eight European countries (Maastricht, The Netherlands; Copenhagen, Denmark; Cambridge, UK; Heraklion, Greece; Potsdam, Germany; Pamplona, Spain; Sofia, Bulgaria; and Prague, Czech Republic). Supplementary Fig. 19 displays the flowchart for DIOGenes participants' selection. Participants followed an initial 8 weeks weight loss intervention on a low-calorie formula diet (LCD 3300 kJ/day, ca. 800 kcal; Modifast; Nutrition et Santé). The mean weight loss during LCD was 10.3 kg (interquartile range: 8.7–12.8 kg)[14].

**Protein measures**. Protein expression was measured using a multiplexed aptamer-based proteomic technology developed by SomaLogic Inc (Boulder, CO). The SOMAscan proteomic assay has been described in previous publications[54,55]. Briefly, this approach uses fluorescently labeled poly-nucleotide aptamers that recognize specific protein epitopes, similar to protein antibodies quantified using relative fluorescence on microarrays. A set of 1129 proteins was measured and quantified directly from plasma samples under fasting conditions before and after intervention. Therefore, resulting protein measurements were reported in relative fluorescence units. Outliers were identified based on interquartile range (IQR;

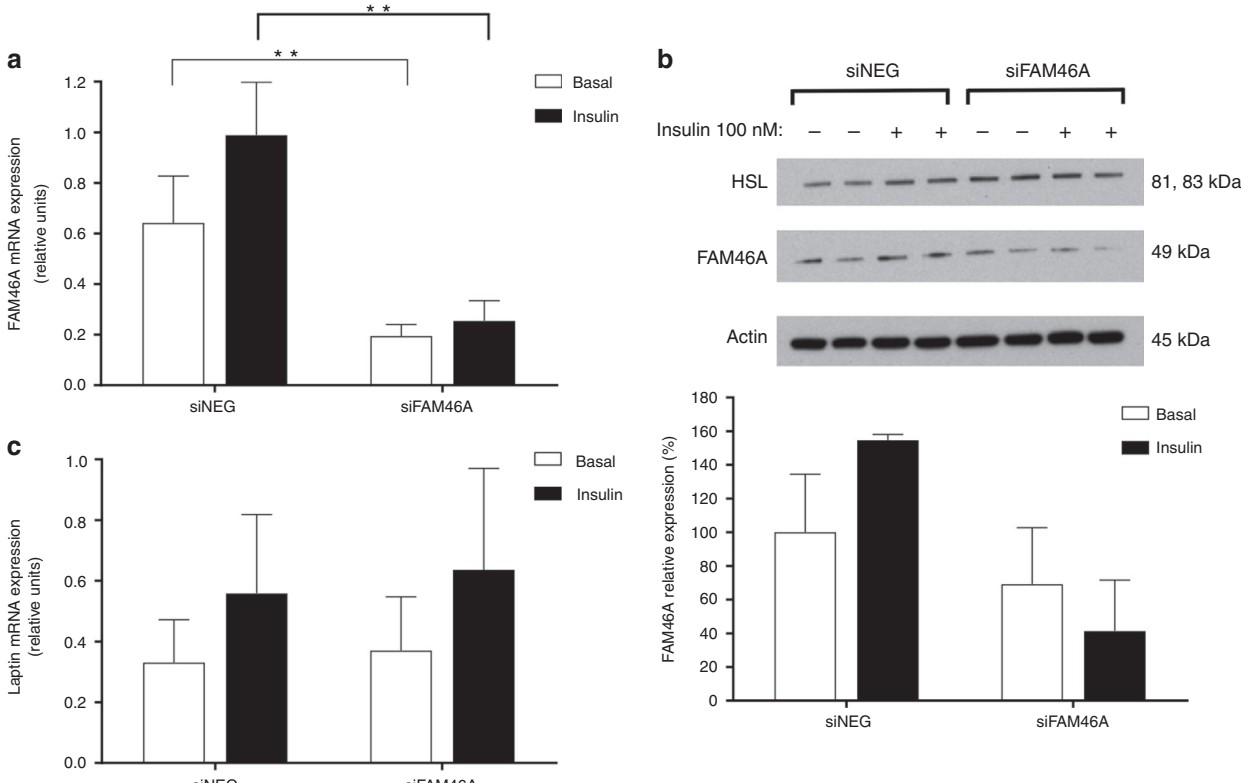

**Fig. 4** *FAM46A* gene silencing effect on leptin protein expression in SGBS adipocytes. SGBS cells were transfected with negative control siRNA (siNEG) or FAM46A-specific siRNA (siFAM46A) and experiments were performed 48 h after transfection. Gene expression and protein measurements for *FAM46A* were conducted at the basal and insulin stimulated states at the same time as leptin expression/secretion analyses at day nine. **a** Gene silencing of *FAM46A* in SGBS adipocytes. Gene expression levels are expressed relative to *TBP* (TATA Binding Protein) gene expression for *FAM46A*. Results are given as the mean ± s.d. (standard deviation) for basal or insulin stimulated state (*n* = 9); **b** Western blot for FAM46A protein expression in SGBS adipocytes in wildtype and *FAM46A* knockdown. *FAM46A* expression was detected in cell lysate by Western blotting. Protein expression levels are expressed relative to Actin expression (*n* = 2). **c** Effects of *FAM46A* knockdown on leptin gene expression at basal non-insulin stimulated conditions and after stimulation with 100 mM insulin. Gene expression levels are expressed relative to *TBP* gene expression. Results are given as the mean ± s.d. (*n* = 9). *$p < 0.05$, **$p < 0.01$ are provided for statistical significance calculated using a Student's *t* test

IQR = Q3–Q1, where Q3 is the third quartile and Q1 is the first quartile of data distribution). Protein data were available for 530 DIOGenes participants before and after the intervention. Extreme outliers were set as missing if out of protein values ± 6 IQR. Participants were excluded (*n* = 18) if more than 5% of proteins were outside ± 4 standard deviations at baseline or after LCD. Protein data were log2 transformed for normality for 512 DIOGenes participants. Principal component analysis did not highlight effects of age, gender or center on protein variation during the intervention. The three first components explained up to 60% of the variance, respectively 49%, 8%, and 3%.

At the time of the analysis, 1,129 proteins were available with the Somalogic Assay. A revision was published recently by the company removing five reagents from future assay results reporting: Alkaline phosphatas, C1s subcomponent, Desmoglein-2, Reticulon-4, and Tumor necrosis factor receptor superfamily member 25. Results from these proteins were kept in the current analysis but not interpreted in the manuscript.

**Gene expression**. Abdominal subcutaneous adipose tissue samples were obtained before and after LCD intervention in subjects following an overnight fast (see Armenise et al.[56] for more information on biopsies and RNAseq analyses). RNA was quantified with Ribogreen (Life Technologies) and RNA quality determined using a Bioanalyzer (Agilent Technologies). TruSeq RNA Library Preparation Kit v2 was used to prepare the sequencing libraries from 250 ng total RNA and quantified by Picogreen (Life Technologies). Fragment sizes were assessed using DNA High Sensitivity Reagent kit on a LabChip GX (Perkin Elmer). A v3 paired-end sequencing flow cell (Illumina) was then used to pool and cluster the libraries at a concentration of 8 pmol. Finally sequencing was performed on a HiSeq 2000 for 2 × 100 cycles following Illumina's recommendations. A first PCA was performed on all data for outlier detection based on exclusion of samples with principal component 1 and 2 out of twice the interquartile range at each time point. RNAseq data included count information for 54,035 genes. Only subjects with

transcriptomics information at CID1 (baseline) and 2 (after LCD intervention) and no outliers based on PCA were kept for further analysis.

Count data were filtered and transformed to normality for analysis. Data were normalized by calculating the reads per kilobase per million mapped reads[57] (RPKM) for each gene. RPKM normalized read counts by library size. Expression matrices were derived from RPKM values at the gene level. Data were log2 transformed with a starting value (prior count = 1, i.e., a starting value used to offset small counts) that was added to all observed counts to avoid missing values when computing logarithms on zero counts. More than 15,000 genes with no read counts at CID1 and 2 and were excluded.

Gene filtering was decided based on more than 50% of participants with more than one read count to keep a maximum of genes in the analysis (i.e., exclusion of genes with 0 read counts in more than 50% of the cohort = detected in more than 50% of the cohort). A list of 26,622 genes fulfilled the selection criteria.

**Genotyping**. Before genotyping, a quality check of genomic and amplified DNA samples was conducted. Samples were then quantified and normalized to approximately 100 ng and 2.0 mg. Genotyping of 748 DIOGenes participants was performed using an Illumina 660W-Quad SNP chip on the Illumina iScan Genotyping System (Illumina, San Diego, CA, USA) in accordance with manufacturer's protocols. The integrated mapping information is based on NCBI's build 37. 498,233 SNPs were genotyped with call rate > 98% and Hardy–Weinberg equilibrium $p > 1.0 \times 10^{-6}$. Imputation was performed on the Michigan Imputation Server[58] using a European 1000 Genomes set reference panel. Imputation data was converted to best-guess genotypes (genotype probability threshold 0.8). Imputed genotypes with an imputation quality score <0.8 and a minor allele frequency < 5% were discarded. A total of 4,020,654 SNPs were available for analysis in 494 participants with proteomics data and a subset of 151 subjects with gene expression data before and after intervention. A description of the two cohorts is provided in Supplementary Table 2.

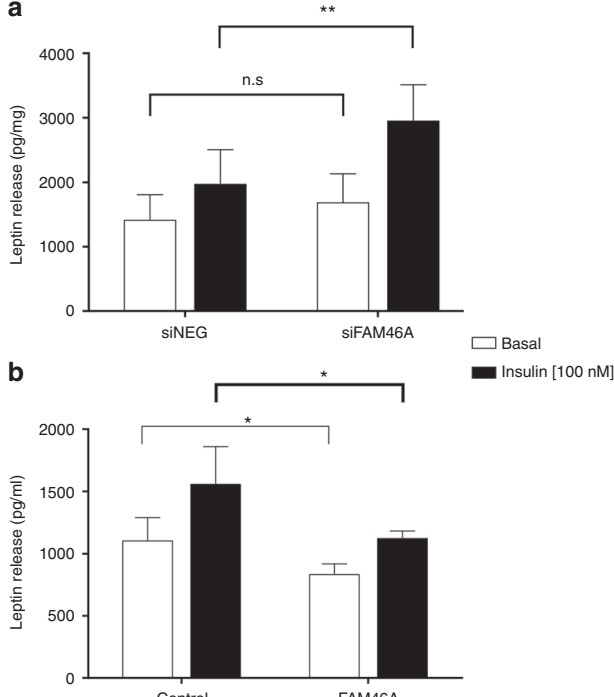

**Fig. 5** Effects of *FAM46A* alterations (knockdown and overexpression) on leptin secretion. **a** leptin release in the *FAM46A* knockdown. **b** leptin release upon overexpression of *FAM46A*. Nine-day-old differentiating preadipocytes were cultured in serum-free basal medium without insulin or after stimulation with 100 nM insulin for 6 h. Conditioned medium was removed and the secreted leptin content was measured using a DuoSet ELISA kit. Accumulated leptin was normalized to total amount of protein from each well. Results are given as the mean ± s.d. ($n = 11$) with *$p < 0.05$, **$p < 0.01$ for statistical significance calculated using a unilateral Student's $t$ test for an increased or a decreased leptin release according to a knockdown or an overexpression of *FAM46A*, respectively

**Proteome-wide association study**. LCD-induced change was quantified by weight loss expressed as fold change in BMI before (CID1) and after (CID2) LCD intervention. Protein change between CID1 and CID2 was computed like BMI change as the $\log_2$ fold change between CID2 and CID1. Association between each protein and BMI was performed using a linear regression adjusted for age, gender, and center. To identify proteins associated with BMI level in obese participants, a first analysis was performed using data at baseline. Then, we conducted a second analysis using proteins and BMI level change between CID1 and CID2 in order to identify proteins associated with weight loss. $p$ values were corrected for multiple testing using Benjamini–Hochberg standard false discovery rate correction[48]. A 5% $p$ value cutoff was used to define the association as significant. Despite a log2 transformation, protein distribution may still have a non-normal distribution, which may imply non-normal residuals and affect association results. The same analysis was performed using a rank-based inverse normal transformation which forces normality of protein distribution based on rank of expression values. $p$ values from both analyses were compared and did not highlighted differences.

**Correlation and network and canonical pathway analysis**. Correlation between proteins was presented in a heatmap. Protein residuals were obtained regressing corresponding expression on age, center, and gender. Protein residuals pairwise correlation was quantified by mean of the Spearman $\rho$ correlation coefficient. Results were plotted in a symmetric heatmap matrix using R function *heatmap.2*.

IPA (Ingenuity Systems, Redwood City, CA) was used to create molecular networks based on protein-coding genes name, expression and $p$ values for each protein available in our data and integrative information from a large structured collection of observations with nearly 5 million findings manually curated from the biomedical literature or integrated from third-party databases[59]. Proteins were clustered into canonical pathways and $p$ values determining an enrichment of selected proteins in these pathways computed using a Fisher's exact test as proposed by IPA. A Benjamini–Hochberg correction for multiple testing was applied and only pathways that met the $p$ value cutoff $p < 0.05$ were considered. Default parameters were used limiting source of data from mammalian studies in the Ingenuity Knowledge Base. The reference set of genes to consider for test $p$

value computation was limited to genes coding for proteins available on the proteomics assay.

**QTL mapping**. pQTLs were mapped using a linear mixed model (LMM) with GCTA software[60]. Sex, age, and center were added to the model as confounding variables with a random polygenic term modeling genome-wide sample structure (the genetic background of all patients in the study). To avoid confounding effects of BMI, pQTL GWAS were performed with and without addition of BMI as confounding factor. GCTA software was used for LMM computation with the 'loco' option that excludes all SNPs belonging to the same chromosome as the SNP under study avoiding multi-collinearity. For each pQTL analysis, the genomic inflation factor (GIF) was estimated using the *estlambda* function available in the GenABEL R package[61]. At baseline, all but one pQTL analysis (pQTL analysis for tPA protein which did not provide any significant results) displayed GIFs ranged from 0.99 to 1.07 (Supplementary Fig. 20a) suggesting that population stratification (if any after correction for population genetic structure using GCTA) was limited. During LCD, GIFs ranged from 0.98 to 1.08 (Supplementary Fig. 20b) leading to the same conclusion.

Correction for multiple testing was performed using SLIDE[62] (Sliding-window method for Locally Inter-correlated markers with asymptotic Distribution Errors corrected). SLIDE is based on a multivariate normal distribution to provide an efficient multiple testing correction which is much faster than the gold standard permutation test with similar accuracy[62] and more tractable when applied to association between 1,129 proteins and millions of common variants. For each gene, *cis*-acting SNPs were located within ±500 Kb of the gene and *trans*-acting outside this region. Functional annotation of SNPs was performed using SNPnexus tool[63] based on NCBI RefSeq database. We first limited our analysis to SNPs available on the Illumina chip used for genotyping with a minor allele frequency (MAF) > 0.05. $p$ values were corrected for multiple testing. SNPs with corrected $p$ value below 5% were then further investigated enriching signals in targeted regions with association results from imputed SNPs.

Enrichment analyses of pQTL SNPs among regulatory elements were performed by two-tailed Fisher's exact test with the following 2 × 2 table: columns; pQTL SNPs and non pQTL SNPs, rows; SNPs within and not within the regulatory element following Takata et al.[64]. Adjustment for multiple testing was performed using Bonferroni correction to account for the number of functional type observed in the study for *cis*-acting and *trans*-acting pQTL separately. Enrichment in regulatory elements could only be performed for baseline pQTL results, the number of pQTLs upon LCD being too low for any accurate comparison.

Transcriptomics data are available only for a subset of DIOGenes participants. Because of statistical power limitation, we could not perform a powerful comparative analysis of eQTL and pQTL results like in Chick et al.[13]. We decided to use gene expression information to confirm the regulatory role of genes coding for proteins with proximal pQTL and identify genes whose expression was likely involved in regulation of proteins with distant pQTL. For each pQTL association signal, genes with available expression data located within 2 Mb around the best associated SNP were identified as potentially regulated candidate genes. eQTL association results for the genetic region using all available common genetic variants were extracted and prioritized based on $p$ values to identify likely regulated genes as the gene with the strongest eQTL signal.

We used the Genotype-Tissue Expression (GTEx) database to investigate whether identified pQTL SNPs were also associated with the expression levels of contiguous genes in different tissues[65]. Basically, a SNP was compared with reported *cis*-eQTL SNPs from GTEx database. Investigation was limited to SNPs with *cis*-eQTL $p$ values below or equal to gene-specific nominal $p$ value thresholds as defined by Ardlie et al.[65] for adipose tissues (subcutaneous with data available from 298 samples with donor genotype and visceral omentum tissues with data available from 185 samples with donor genotype).

pQTL SNPs were evaluated for association with BMI in the Locke et al.[2] published large GWAS. To assess the significance of the n observed pQTL SNPs, we used a resampling approach where n SNPs were drawn at random, and we counted how many of those were associated with BMI (with unadjusted alpha 5%). By repeating this process 1,000 times we built the null distribution of enrichment and derived the empirical $p$ value of enrichment computing the proportion of sampled SNP lists with a higher number of nominal association with BMI in the GWAS.

**SGBS adipocyte cell culture**. All the chemicals were from Sigma-Aldrich (Saint Louis, MO, USA) unless stated otherwise. SGBS adipocytes were obtained from Prof. Wabitsch at the University of Ulm, Germany. These preadipocyte cells are derived from adipose tissue of a patient with Simpson–Golabi–Behmel syndrome (SGBS) and have a unique retained capacity for adipogenic differentiation[66]. The SGBS cultures have been repeatedly shown to be mycoplasma-free and negative for the pathogens HIV, hepatitis B, and EBV. Cells were cultured in basal medium (DMEM/F12 medium containing 1% PS and 33 µM biotin, 17 µM panthonenate, and 10% FBS) at 37 °C. To obtain basal serum-free medium (OF), supplement of FBS was omitted. Differentiation was induced at near confluence by incubation with OF medium supplemented with 10 µg/ml apo-transferrin, 20 nM human recombinant insulin, 100 nM hydrocortisone, and 0.2 nM tri-iodothyroxine (T3). During the first 4 days OF medium was additionally supplemented with 25 nM

dexamethasone, 250 μM 1-methyl-3-isobutylxanthine (IBMX), and 2 μM rosigli-tazone (Cayman Chemical).

**FAM46A gene silencing in differentiating SGBS**. Silencing of the *FAM46A* gene was achieved by RNA interference through electroporation. Briefly, on day 7 of differentiation, SGBS cells were detached with trypsin/EDTA and counted. Silencer Select Negative Control siRNA (siNeg) or gene-specific siRNA targeting human *FAM46A* (Life Technologies, Carlsbad, USA) were delivered into adipocytes at a concentration of 25 nM per 400,000 cells with the Neon Transfection System (Invitrogen, Madison, WI, USA) using the following parameters: 1100 V, 20 ms, 1 pulse. After transfection cells were incubated for 48 h in the same media.

**FAM46A overexpression**. Overexpression of the *FAM46A* gene was achieved by cDNA transfection through electroporation. Briefly, on day 7 of differentiation, SGBS cells were detached with trypsin/EDTA and counted. pCMV6 empty vector (Control) or *FAM46A* (NM_017633) full human cDNA ORF Clone (OriGene, Rockville, MD, USA) were delivered into adipocytes at a concentration of 0.5 μg per 400,000 cells with the Neon Transfection System (Invitrogen, Madison, WI, USA) using the following parameters: 1100 V, 20 ms, 1 pulse. After transfection cells were incubated for 48hr in the same media.

**Gene expression measurements for FAM46A and leptin and PPARγ**. Total RNA from SGBS adipocytes was isolated using the Rneasy Plus Mini kit (Qiagen) at day 9. After assessing quantity and quality with a Nano-Drop, 0.5 μg total RNA was reverse transcribed using the High Capacity cDNA Reverse Transcription kit (Applied Biosystems, Foster City, CA, USA) according to manufacturer's instructions.

Expression levels of mRNA were determined using a SYBR Green kit LightCycler 1536 DNA Green Master kit using the LightCycler Instrument (Roche, Basel, Switzerland). Oligo sequences were designed using the Biology Workbench website[67]. Primers for human sequences were obtained from Life Technologies Europe. *FAM46A, Leptin, CEBPα,* and *PPARγ* mRNA levels were normalized against TATA box-binding protein (TBP). Primers used as listed in Supplementary Table 3.

**Protein quantification**. At day nine of differentiation SGBS cells were homo-genized in ice cold M-PER Mammalian Protein Extraction Reagent (#78501, Thermo Fisher Scientific, Waltham, MA, USA) supplemented with 250 mM Sucrose, 1 mM EDTA, 1% NP-40 substitute and complete protease inhibitor cocktail (Roche GmbH, Mannheim, Germany). After incubation on ice for 10 min and centrifugation at 10,000 × g for 10 min at 4 °C, the middle phase was removed and proteins were quantified using the Bio-Rad *DC* Protein Assay Kit. SDS-PAGE followed by Western blotting was performed using antibodies against FAM46A (1:1000 dilution, PA5-23898, Thermo Fisher Scientific, Waltham, MA, USA) or HSL (1:2000 dilution, #4107, Cell Signalling Technology, Danvers, MA, USA) following manufacturer's standard protocol and ß-Actin (1:4000 dilution, #3700, Cell Signalling Technology, Danvers, MA, USA) was used for normalization. Species appropriate secondary antibodies conjugated to horseradish peroxidase (1:10,000 dilution, anti-mouse 711-035-152 or anti rabbit 715-035-150, Jackson ImmunoResearch Laboratories, Inc, West Grove, PA, USA) were used and proteins visualized and quantified by enhanced chemiluminescence using the Bio-Rad GS-900 calibrated Densitometer.

**Leptin and adiponectin secretion**. Leptin and adiponectin concentrations were measured at day 9 in fully differentiated SGBS cells. Cells were washed two times with serum-free Basal medium and left untreated or treated with 100 nM insulin. After 6 h of incubation, aliquots of the medium were removed and frozen at −20 °C for leptin or adiponectin measurements respectively.

Leptin and adiponectin secretion were measured using the human leptin or adiponectin DuoSet ELISA Development kits (R&D Systems, Inc., Minneapolis, MN, USA, sensitivity: 0.031 ng/ml for leptin and 0.062 ng/ml for adiponectin respectively). The assays were conducted in 96-well microplates and read out on a SpectraMax i3 spectrophotometer (Molecular Devices, LLC, Sunnyvale, CA, USA). The absorbance value was measured at 450 nm and normalized to the protein concentration measured by the Bio-Rad *DC* Protein Assay.

**Software**. General statistical analysis was performed using R statistical environment version 3.3.1. We used LocusZoom software for plotting regional association plots and LD with lead SNP using the 1000 genome CEU population data (hg19/1000 Genomes Mar 2012 EUR).

**URLs**. Biology Workbench http://workbench.sdsc.edu;
GCTA, http://cnsgenomics.com/software/gcta/;
https://imputationserver.sph.umich.edu/index.html#!pages/home;
GTOOL, http://www.well.ox.ac.uk/~cfreeman/software/gwas/gtool.html;
GTEx database, http://www.gtexportal.org/home/;
IPA, http://www.ingenuity.com/;
Michigan Imputation Server, https://imputationserver.sph.umich.edu/;

LocusZoom, http://locuszoom.org/

**Data availability**. Gene expression data are available from the Gene Expression Omnibus under accession GSE95640. Any additional data (beyond those included in the main text and Supplementary Information) that support the findings of this study are available upon reasonable request by contacting the corresponding author at jerome.carayol@rd.nestle.com.

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

## Acknowledgments

We are grateful to the Centre National de Génotypage, Evry, France for RNAseq data generation, to M. Wabitsch, University of Ulm, Germany for providing the SGBS cells and to Margharita Springer for reviewing the manuscript. The DIOGenes trial was funded by the European Commission, contract no. FP6-2005-513946. The funding source had no role in the study design, data collection, data analysis, data interpretation, or writing of the report. RNAseq analysis was supported through ESGI funding from the European Union Seventh Framework Programme (FP7/2007–2013) under grant agreement no. 262055.

## Author contributions

J.C. and J.H. designed, supervised the study, interpreted results and wrote the manuscript with input from all authors. J.C. performed statistical analyses. W.S. and A.A. designed the DiOGenes clinical study, C.A., A.D.C., and J.C. performed protein data processing and quality control; G.L., P.D., D.L., and N.V. performed expression data generation. A. V. supervised omics data processing, Q.C. and integration, processed genotype data and helped with statistical analysis. S.M., C.C., and P.D. performed genotype data generation. C.C. performed SGBS functional analyses. J.C. had primary responsibility for final content.

## Additional information

**Competing interests:** J.C., A.V., G.L., C.C., S.M., P.D., and J.H. are full-time employees at Nestlé Institute of Health Sciences SA. C.A. and A.D.C. are full-time employees at Quartz Bio SA. W.H.M.S. is medical consultant for N&S and is an unpaid scientific advisor for the International Life Science Institute, ILSI Europe. W.H.M.S. reports having received research support from several food companies such as Nestlé, DSM, Unilever, Nutrition et Santé and Danone as well as Pharmaceutical companies such as GSK, Novartis and Novo Nordisk. A.A. reports grants and personal fees from Global Dairy Platform, USA; McCain Foods, USA; McDonald's, USA; Arena Pharmaceuticals Inc, USA; Basic Research, USA; Dutch Beer Knowledge Institute, NL; Gelesis, USA; Novo Nordisk, DK; Orexigen Therapeutics Inc., USA; S-Biotek, DK; Twinlab, USA; Vivus Inc.,

USA; and grants from Arla Foods, DK; Danish Dairy Research Council, Nordea Foundation, DK outside the submitted work; and Royalties received for the book first published in Danish as "Verdens Bedste Kur" (Politiken, Copenhagen), and subsequently published in Dutch as "Het beste dieet ter wereld" (Kosmos Uitgevers, Utrecht/Antwerpen), in Spanish as "Plan DIOGenes para el control del peso. La dieta personalizada inteligente" (Editorial Evergráficas, Léon), and in English as "World's Best Diet" (Penguin, Australia).

