## [Peer Review File · Nature Communications]

Reviewer #1 (Remarks to the Author):

The authors have studied associations of 1,129 plasma proteins with BMI in 494 obese individuals before and during an 8-week weight loss intervention, and subsequently examine the associations of genome-wide variants with the levels of the proteins found associated with BMI. They find 192 proteins associated with BMI at baseline and 104 proteins associated with BMI change during the intervention. They identify 55 cis and trans-pQTLs for the proteins associated with BMI at baseline, and 3-trans-pQTLs for the proteins associated with BMI change. Using adipose tissue RNAseq expression data from a subset of 151 participants, and publically accessible data from the GTEx Consortium, they compare the overlap between eQTLs and the identified cis and trans-pQTLs. In siRNA knockdown studies in mice, they experimentally validate FAM46 as a novel trans-regulator for leptin levels. The results and study design are novel and interesting; to my knowledge, there are no previous genome-wide association studies of proteomic changes during a lifestyle intervention.

MAJOR COMMENTS

1. In the Abstract and Introduction, the authors emphasize the importance of addressing the question of how much of the observed variation in protein levels is driven by genetic variation. However, they do not seem to directly address this question in the manuscript, and their conclusion "we showed that one third of 192 proteins associated with BMI at baseline are under genetic control" seems only based on the finding that statistically significant cis or trans-pQTLs were found for 55 out of 192 BMI-associated proteins. To address this point directly, testing for the narrow-sense heritability in protein levels using GCTA or other software would be warranted.
2. The authors have performed genetic association tests to search for pQTLs for proteins associated with BMI either at baseline or during the weight loss intervention. The resulting SNPs may be potential candidate SNPs for adiposity. Thus it would be very relevant to look-up data in publically accessible GWAS meta-analysis results to see whether the identified pQTLs are associated with adiposity-traits. In addition, it might be relevant to perform an enrichment test to find out whether the identified pQTLs are more likely to be associated with BMI as compared to all other SNPs. Vice versa, it might be informative to look-up the associations of known BMI-associated loci with protein levels.
3. The authors find a pQTL for changes in leptin levels in response to intervention. It would be relevant to perform a look-up for this pQTL in the publically accessible GWAS results for leptin levels.
4. The comparisons between pQTL and eQTL results are in part hard to follow. The authors are sometimes referring to eQTL results from their individual-level data of 151 participants and sometimes to eQTL results from the GTEx consortium. However, it does not always become clear from the text which data they are referring to. Please be more clear about this aspect. Please also report how many individuals were included in the GTEx consortium lookups and which tissue type each result is based on.

MINOR COMMENTS

1. From the manuscript it does not become clear whether the authors used fasting or non-fasting plasma. Please clarify this point.
2. Page 6: Was there an enrichment of the cis- and trans-pQTLs in any particular genomic element (intergenic, coding, intronic etc.) as compared to all other SNPs?
3. Pages 9-10: The following two sentences contradict: "Of the 42 proteins associated to BMI both

at baseline and during LCD, 14 displayed significant pQTL associations at baseline but none reached significant level during the LCD intervention LCD after correcting for multiple testing.” and “Only one protein among the 42 proteins that showed a correlation with BMI both at baseline and during the LCD, displayed a pQTL association signal during LCD: leptin.”

Reviewer #2 (Remarks to the Author):

This paper addresses the question of how much of the observed variation in protein levels in obesity is driven by genetic variation. The study is based on samples from 494 obese subjects that followed an 8 weeks weight loss intervention on a low calorie formula diet. Aptamer-based proteomics data for 1,100 proteins was collected from blood plasma before and after intervention and 1000G imputed genotype data was used in the genetic association study. Gene expression was determined using RNAseq in samples from adipose tissue biopsies in a smaller subset of 151 participants. The authors identified 55 BMI-associated pQTLs at baseline and 3 pQTLs after the intervention, and singled out FAM46A as a new trans regulator for leptin.

This paper addresses an important question and provides original new data. However, there are several issues with the data analysis and the experimental confirmation. Without the following information the validity of the results is hard to assess:

Full statistics for the association of protein expression and protein expression change with BMI should be provided and extensively discussed (not merely presented as volcano plots).

Similarly for the genetic associations: effect size, effect allele and directionality need to be reported.

For the three key associations, individual data points/ bee plots / box plots by genotype should be presented.

Report the level of genomic inflation in the pGWAS.

Since this study does not contain a replication, results should be discussed in the light of related previous studies using the same technology, i.e. previously published pQTL associations (PMID 22595970 Pathways in Cardiovascular Disease and Identification of cis-regulatory variation influencing protein abundance levels in human plasma and

<http://biorxiv.org/content/early/2016/11/09/086793>). What replicates? What is different?

Similarly, protein associations should be put in context of a recent study on cardiovascular risk traits (PMID 27444932 Aptamer-Based Proteomic Profiling Reveals Novel Candidate Biomarkers).

The experimental data supporting the FAM46A gene silencing (Figure 4) is not very convincing. Fig 4a does not show any error bars – is this based on a single experiment? What does “Results are given as the +/- SD of two independent experiments” mean? Was the SD computed based on N=2? Individual data points should be provided to assess the variability of the observations, and more biological replicates are needed, ideally using complementary techniques. Overall I feel that the presented experimental data is not very convincing and requires more backing.

What background gene set was used in the IPA network/gene set enrichment analysis? How was multiple testing accounted for? P-values on the order of 1E-4 are typically seen in IPA analyses and may well be false positives. Also, the plotted networks provided as supplemental data are not very helpful as there is no information on the supporting data (which can only be accessed in an active IPA session).

Specific comments:

According to an update by Somalogic, the aptamer that is supposed to target C1s actually Complement Factor H: “Mass spectrometry identifies Complement Factor H as the protein CFH, enriched targets from plasma when the SOMAmer reagent 3590-8 is used for affinity capture.”

See: <http://www.somalogic.com/wp-content/uploads/2017/01/Revision-Anno-uncement-of-the-1.3k-Version-SOMAscan-Assay-v2.pdf>

line 242: duplicated word “LCD”

line 349: “amonacids” should read “amino acids”

line 438: report how many participants were excluded

line 441: report how much of the variance is explained by the first/first three principal components

line 525: why limit IPA to "human studies" ? Most information about protein function comes from mouse knock-out studies

line 534: report the result of performing GWAS with/without BMI as confounder

Reviewer #3 (Remarks to the Author):

The manuscript by Carayol et al reports a large-scale protein quantitative trait locus (pQTL) analysis based on a predetermined set of 1,129 plasma proteins from 494 obese subjects before and after a weight loss intervention. The study identified 55 BMI-associated cis- and trans- pQTL's at baseline and 3 trans-pQTLs after a 6 week 800 caloric diet (LCD) intervention. The study identified strong cis- or trans- pQTLs for proteins that have been previously associated with obesity, including two inflammation-related genes (IL-1 R AcP and IL-16), sE-Selectin and leptin. LCD intervention activated primarily trans-pQTLs. A novel trans-pQTL near FAM46A for leptin after LCD intervention was identified. The pQTL, however, had no detectable effect on leptin expression at baseline. In vitro study showed that knockdown of FAM46A increased insulin-stimulated leptin secretion in SGBS human adipocytes.

Major points:

1. The timing of the acquisition of the plasma samples relative to meals should be stipulated. As should the weight status of the subjects: that is, in the LCD portion of the study were the subjects still in negative energy balance? A schematic of the clinical protocol in this regard would be helpful.
2. Additional details regarding the acquisition of adipose tissue for expression analyses should be given, including weight status relative to the LCD and sites analyzed.
3. How does FAM46A knockdown affect leptin secretion in non-insulin stimulated SGBS cells? Because the SNP near FAM46A appeared to only affect leptin level after the LCD intervention, when insulin levels are likely to be lower, this set of data would be important to a better understanding of the role of FAM46A in regulating leptin level in LCD condition.
4. Again, considering that the pQTL SNP next to FAM46A had no association with plasma leptin concentrations at baseline and that FAM46A expression is increased after low caloric diet intervention, the in vitro confirmation study should also examine the effect of FAM46A overexpression on basal and insulin-stimulated leptin expression and secretion in SGBS cells.
5. What were the directions of the effect of eQTL and pQTL SNPs on FAM46A expression and plasma leptin concentrations? Were they directionally consistent with the proposed regulatory role of FAM46A on leptin production/secretion?
6. Do the authors have opinions regarding why PUR8, ferritin and CA1 levels are correlated with the same pQTL?
7. TIPARP was suggested to be the gene influencing the pQTL that affects ferritin and potentially leptin concentration through iron-ferritin metabolism. As indicated by the authors, a previous GWAS has identified TIPARP as a modulator of circulating leptin concentrations independent of BMI. Why was the TIPARP signal not detected as a significant pQTL for leptin in this study?
8. Why was adiponectin, a protein produced in large amount by adipose tissue with plasma concentrations inversely related to adiposity not reported here, regardless of changes seen.
9. A recent paper looking for genes modifying circulating leptin concentrations should be consulted for possible implication of FAM46A. And should in any event be mentioned in the discussion re strategies for finding such genes. Kilpeläinen, Tuomas O., et al. "Genome-wide meta-analysis uncovers novel loci influencing circulating leptin levels." *Nature communications* 7 (2016).

Minor points:

1. In "Result" and Discussion", it is not clear how the pQTL SNP in CPN1 for Calpastatin is related to an eQTL for CWF19L1? Is there any known or inferred functional connection between calpastatin and CWF19L1?
- 2.
3. What was the rationale for use of the X cell line, from individuals known to have primary
4. Some of the gene symbol "ABO" looked like "AB0". Was there a typo? Disturbances in cell growth?

Dear Editor,

Thank you very much for your editorial review and sharing the comments from the three reviewers. Their respective comments have been very useful and we believe it has improved the quality of our manuscript.

We include below a point-by-point response to the reviewers' comments.

Reviewer #1 (Remarks to the Author):

The authors have studied associations of 1,129 plasma proteins with BMI in 494 obese individuals before and during an 8-week weight loss intervention, and subsequently examine the associations of genome-wide variants with the levels of the proteins found associated with BMI. They find 192 proteins associated with BMI at baseline and 104 proteins associated with BMI change during the intervention. They identify 55 cis and trans-pQTLs for the proteins associated with BMI at baseline, and 3-trans-pQTLs for the proteins associated with BMI change. Using adipose tissue RNAseq expression data from a subset of 151 participants, and publically accessible data from the GTEx Consortium, they compare the overlap between eQTLs and the identified cis and trans-pQTLs. In siRNA knockdown studies in mice, they experimentally validate FAM46 as a novel trans-regulator for leptin levels. The results and study design are novel and interesting; to my knowledge, there are no previous genome-wide association studies of proteomic changes during a lifestyle intervention.

Response: Thank you for your interest in our work. We agree that we're not aware of any other large pQTL studies for specific clinical outcomes in humans.

MAJOR COMMENTS

1. In the Abstract and Introduction, the authors emphasize the importance of addressing the question of how much of the observed variation in protein levels is driven by genetic variation. However, they do not seem to directly address this question in the manuscript, and their conclusion "we showed that one third of 192 proteins associated with BMI at baseline are under genetic control" seems only based on the finding that statistically significant cis or trans-pQTLs were found for 55 out of 192 BMI-associated proteins. To address this point directly, testing for the narrow-sense heritability in protein levels using GCTA or other software would be warranted.

Thank you for this comment. By "genetic control", we meant to identify how many proteins would potentially be regulated by one or more QTLs. We clarified the corresponding sentences in the Abstract and the Introduction (page 2 and 4 respectively: "*The role of genetic variation in determining protein levels variation has not been assessed in obesity*") then in the Conclusion (page 13: "*We showed that one third of 192 proteins associated with BMI at baseline are likely regulated by common genetic variants*").

We also agree with the reviewer that narrow-sense heritability (h^2) in proteins level would be valuable information. Using GCTA, we estimated h^2 on both baseline protein levels and their fold-changes upon LCD. However, these h^2 estimates have very large variance, preventing any firm conclusion. Such large variance was expected given the type of study (cohort of unrelated subjects) and moderate sample size (as expected from clinical intervention). Generally speaking robust h^2

values are obtained from twin studies. GCTA and other similar tools were developed to derive an approximate estimate of heritability. Few hundred pairs of twins are required to estimate moderate h^2 using the twin design. By contrast, because GCTA like methods attempts to extract small signals of genetic similarity from the noise of hundreds of thousands of SNPs, samples of thousands of subjects are required (Plomin and Deary. Mol Psych. 2015. Genetics and intelligence differences: five special findings; [10.1038/mp.2014.105](https://doi.org/10.1038/mp.2014.105)).

We do not have the required sample size for accurate h^2 estimates and therefore we decided not to include these results in the paper.

2. The authors have performed genetic association tests to search for pQTLs for proteins associated with BMI either at baseline or during the weight loss intervention. The resulting SNPs may be potential candidate SNPs for adiposity. Thus it would be very relevant to look-up data in publically accessible GWAS meta-analysis results to see whether the identified pQTLs are associated with adiposity-traits. In addition, it might be relevant to perform an enrichment test to find out whether the identified pQTLs are more likely to be associated with BMI as compared to all other SNPs. Vice versa, it might be informative to look-up the associations of known BMI-associated loci with protein levels.

This is a valid point, we compared our pQTL results with those from the largest meta-GWAS performed on BMI by Locke et al. (2015). A section entitled **“Comparison of pQTLs with known GWAS loci”** was added to the manuscript in the Results section (page 12):

“Locke et al.² performed the largest GWAS of BMI in up to 339,224 individuals. Among all pQTL SNPs identified in our study, 16 were nominally (pvalue < 5%) associated to BMI in the published GWAS² (Supplementary Table 4). Overall the identified pQTL SNPs were significantly enriched in SNPs nominally associated to BMI in Locke et al.² (permutation test pvalue estimated to 0.009). No LCD pQTL displayed association to BMI in the published GWAS². Enlarging the comparison to common variants in LD with pQTL SNPs did not extract more BMI association results. Inversely, pQTL signals from the 97 BMI-associated SNPs identified on Locke et al.² were extracted from the current study. Information from 62 SNPs was available. Missing SNPs were replaced by SNPs in LD ($r^2 > 0.8$) for 21 SNPs leading to 83 SNPs with pQTL results. For each protein, pQTL association results for this set of SNPs were extracted and pvalues corrected for multiple testing using Benjamin-Hochberg method⁶⁷.

Thirty-one out of 83 GWAS SNPs displayed nominal association with expression levels from one protein at least at baseline (Supplementary Table 5). All were identified as trans-regulating pQTLs including pQTLs for FTO and CRP. During LCD, 16 SNPs were identified as potential pQTLs (Supplementary Table 6). However, at the proteome-wide level used in our study no SNP from Locke et al.² GWAS reached significance as a pQTL at baseline or during LCD.

A description of the permutation process for pvalue computation is provided in the Method section (page 23):

“pQTL SNPs were evaluated for association to BMI in Locke et al.² published large GWAS results. To assess the significance of the n observed pQTL SNPs, we used a resampling approach where n SNPs were drawn at random, and we counted how many of those were associated with BMI (with unadjusted alpha 5%). By repeating, this process 1,000 times, we built the null distribution of enrichment and derive the

empirical p-value of enrichment computing the proportion of sampled SNPs lists with a higher number of nominal association to BMI in the GWAS."

3. The authors find a pQTL for changes in leptin levels in response to intervention. It would be relevant to perform a look-up for this pQTL in the publically accessible GWAS results for leptin levels.

Our two leptin pQTL SNPs (i.e. SNPs associated with change in leptin levels upon LCD) did not show any association with leptin levels, in the Kilpelainen et al. GWA results. We found similarly negative results when testing the link between those SNPs and baseline leptin levels in the Diogenes data. Thus our two SNPs (and their neighboring SNPs) only show association with leptin levels in response to the LCD intervention. A paragraph was added to the Discussion to comment this observation (page 15 to 16):

"None of the two leptin associated pQTL SNPs or any SNPs in the surrounding region was found associated to leptin in Kilpelainen et al.⁶ meta-analysis. This study used cohorts recruited from the general population and tested association between common genetic variants and circulating leptin level while we identified pQTL SNPs associated to leptin protein expression change during a weight-loss intervention in obese subjects. This may be due to a specific induction of the FAM46A gene/protein during weight loss. The reciprocal relationship between FAM46A and leptin is also apparent in the shift of gene expression during adipocyte differentiation. Thus the external stimulus and drastic change would be necessary to identify the relationship. We believe that this is a first example that an environmental cue (like a low calorie diet) that will induce large changes in gene and protein expression can lead to the identification of pQTL signals not detectable at baseline homeostasis."

4. The comparisons between pQTL and eQTL results are in part hard to follow. The authors are sometimes referring to eQTL results from their individual-level data of 151 participants and sometimes to eQTL results from the GTEx consortium. However, it does not always become clear from the text which data they are referring to. Please be more clear about this aspect. Please also report how many individuals were included in the GTEx consortium lookups and which tissue type each result is based on.

The manuscript was modified to clarify references coming from our own eQTL results or the GTEx database and the type of tissue the results pertain to. Additional information was provided about the number of samples used in GTEx for each type of tissue (page 23):

"Investigation was limited to SNPs with cis-eQTL pvalues below or equal to gene-specific nominal pvalue thresholds as defined by Ardlie et al⁸ for adipose tissues (subcutaneous with data available from 298 samples with donor genotype and visceral omentum tissues with data available from 185 samples with donor genotype).

MINOR COMMENTS

1. From the manuscript it does not become clear whether the authors used fasting or non-fasting plasma. Please clarify this point.

Blood sampling was made under fasting condition. This information was added to the manuscript page 18:

"A set of 1,129 proteins was measured and quantified directly from plasma sample under fasting condition before and after intervention."

2. Page 6: Was there an enrichment of the cis- and trans-pQTLs in any particular genomic element (intergenic, coding, intronic etc.) as compared to all other SNPs?

We tested enrichment of pQTL SNPs among regulatory elements using a two-tailed Fisher's exact as described in the Method section (page 22):

"Enrichment analyses of pQTL SNPs among regulatory elements were performed by two-tailed Fisher's exact test with the following 2 × 2 table: columns; pQTL SNPs and non pQTL SNPs, rows; SNPs within and not within the regulatory element following Takata et al.⁹⁷. Adjustment for multiple testing was performed using Bonferroni correction to account for the number of functional type observed in the study for cis and trans-acting pQTL separately. Enrichment in regulatory element could only be performed for baseline pQTL results. The small number of pQTLs upon LCD being too low for any accurate comparison.

Results were added to Table 3 and commented in the Results section (page 6):

"Cis-acting pQTL were significantly enriched among variants in 5' upstream (corrected $p=6.1 \times 10^{-4}$) and coding (corrected $p=2.2 \times 10^{-4}$) regions."

3. Pages 9-10: The following two sentences contradict: "Of the 42 proteins associated to BMI both at baseline and during LCD, 14 displayed significant pQTL associations at baseline but none of them reached significant level during the LCD intervention LCD after correcting for multiple testing." and "Only one protein among the 42 proteins that showed a correlation with BMI both at baseline and during the LCD, displayed a pQTL association signal during LCD: leptin."

The sentences were rephrased for clarity pages 9:

"Of the 192 proteins associated to BMI at baseline 42 also showed significant association during LCD. Fourteen (Supplementary Table 3) of these 42 proteins displayed significant pQTL signals at baseline but not after the LCD."

and page 10:

"Among the 42 proteins associated to BMI both at baseline and during the LCD, only leptin displayed pQTL signal during LCD."

Reviewer #2 (Remarks to the Author):

This paper addresses the question of how much of the observed variation in protein levels in obesity is driven by genetic variation. The study is based on samples from 494 obese subjects that followed an 8 weeks weight loss intervention on a low calorie formula diet. Aptamer-based proteomics data for 1,100 proteins was collected from blood plasma before and after intervention and 1000G imputed genotype data was used in the genetic association study. Gene expression was determined using RNAseq in samples from adipose tissue biopsies in a smaller subset of 151 participants. The authors identified 55 BMI-associated pQTLs at baseline and 3 pQTLs after the intervention, and singled out FAM46A as a new trans regulator for leptin.

This paper addresses an important question and provides original new data. However, there are several issues with the data analysis and the experimental confirmation. Without the following information the validity of the results is hard to assess:

We thank you for a very thorough review and your interest in our work. We provide below clarifications to your comments.

Full statistics for the association of protein expression and protein expression change with BMI should be provided and extensively discussed (not merely presented as volcano plots).

We now provide Supplemental Table 1 and 2, containing the full results of our association analysis between proteins and BMI. These analyses were performed as a mean to restrict the number of pQTL analyses. We clarified this accordingly in the text, and we also provide a description of the top results page 5:

"Leptin known as the "satiety" hormone widely investigated in obesity²⁵ displayed the strongest association to BMI at baseline. Other proteins already referenced in the literature as linked to obesity or obesity-related traits included CRP^{26,27}, IGFBP-1²⁸, kalistatin²⁹, factor H³⁰ or antithrombin III³¹. Proteins associated with BMI for the first time in our study included UNC5F4 encoded by UNC5D a receptor for netrin involved in cell migration. Netrins are a class of proteins involved in angiogenesis³². Netrin-1 likely plays a key role in the retention of macrophages in the visceral white adipose tissue during obesity promoting insulin resistance and chronic inflammation³³. (...) On the other hand, some proteins were identified as potentially linked to obesity for the first time like RET proto-oncogene or granulysin. Granulysin protein is present in cytotoxic granules of cytotoxic T lymphocytes and natural killer cells. Lynch et al.³⁸ observed significantly less circulating natural killer and cytotoxic T lymphocytes cells in obese patients compared to lean controls."

Similarly for the genetic associations: effect size, effect allele and directionality need to be reported.

Requested statistical information were added to pQTL results (minor allele, frequency, corresponding coefficient, standard error and pvalue from genetic association analysis with and without adjustment on BMI) at baseline and during LCD in tables 1 and 2 respectively.

For the three key associations, individual data points/ bee plots / box plots by genotype should be presented.

At baseline, boxplots are provided for SNP rs3765964 with CA6 (gustin) and rs10282458 with chemerin protein level in Supplementary Fig. 6. During LCD same boxplots (Supplementary Fig. 8) are provided for rs7512601 with gustin, rs11113832 with chemerin and rs481777 with leptin protein level. Reference to these figures were added in the Results section entitled "**pQTL effects on gustin, chemerin and leptin protein expression**" page 9.

Report the level of genomic inflation in the pGWAS.

Genomic inflation factors were computed for all protein GWAS using the *estlambda* function from "GenABEL" R package. Distribution are provided in histograms (Supplementary Figure 13), results and comments were added to the manuscript in the Method section (page 22):

"For each pQTL analysis, the genomic inflation factor (GIF) was estimated using the estlambda function available in the GenABEL R package⁹². At baseline, all but one pQTL analysis (pQTL analysis for tPA protein which did not provide any significant results) displayed GIFs ranged from 0.99 to 1.07 (Supplementary Fig. 13a) suggesting that population stratification (if any after correction for population genetic structure using GCTA) was limited⁹³. During LCD, GIFs ranged from 0.98 to 1.08 (Supplementary Fig. 13b) leading to the same conclusion."

Since this study does not contain a replication, results should be discussed in the light of related previous studies using the same technology, i.e. previously published pQTL associations (PMID 22595970 Pathways in Cardiovascular Disease and Identification of cis-regulatory variation influencing protein abundance levels in human plasma and <http://biorxiv.org/content/early/2016/11/09/086793>). What replicates? What is different?

Similarly, protein associations should be put in context of a recent study on cardiovascular risk traits (PMID 27444932 Aptamer-Based Proteomic Profiling Reveals Novel Candidate Biomarkers).

Indeed, replication with an external cohort would prove very valuable. However, the DiOGenes study is unique both by design and by size. The clinical intervention represented a budget of 20M euros and a trial over 5 years (excluding any omics analyses). Replication with a similar cohort is desirable but cannot be achieved within the scope of the present work. Comparison with other studies, using similar technology, is possible. Yet, since the tested clinical outcomes are different, the interpretation would prove challenging.

However, since cardiovascular diseases (CVD) are among obesity comorbidities, it would be interesting to compare our results to Ngo et al (2016). As observed in table 14 from their Supplementary Material, many obesity associated proteins were associated to Framingham Risk Score predictor of CVD like adiponectin, SHBG, IGFBP-2, CRP, IGFBP-1, Growth Hormone Receptor, sE-Selectin. Their list also contains pQTL proteins which were deeply investigated in our study like Carbonic Anhydrase 6 (gustin), TIG2 (chemerin), HSP-70 and leptin. Unfortunately we are not able to differentiate from their list proteins linked to obesity from proteins associated to CVD since the regression used to test the association was not corrected for BMI as a surrogate marker of obesity. Few information was made available about the

subgroup of participants but none about weight or BMI. We observed in Supplementary Table 6, a slightly elevated total cholesterol mean (207mg/dl +/- 37) and systolic blood pressure (129mmHg +/- 20).

To address your comment, we added a comparative analysis of our pQTL with results from the largest GWAS performed on obesity by Locke et al. (2015). A new chapter entitled "**Comparison of pQTLs with known GWAS loci**" was added to the manuscript in the Results section (page 11 to 12). We also compared our leptin pQTL identified during LCD to results from a large GWAS performed by Kilpelainen et al. to identify common genetic variants associated to circulating leptin level in the general population (Page 15):

"None of the two leptin associated pQTL SNPs or any SNPs in the surrounding region was found associated to leptin in Kilpelainen et al.⁶ meta-analysis. This study used cohorts recruited from the general population and tested association between common genetic variants and circulating leptin level while we identified pQTL SNPs associated to leptin protein expression change during a weight-loss intervention in obese subjects. This may be due to a specific induction of the FAM46A gene/protein during weight loss."

The experimental data supporting the FAM46A gene silencing (Figure 4) is not very convincing. Fig 4a does not show any error bars – is this based on a single experiment? What does "Results are given as the +/- SD of two independent experiments" mean? Was the SD computed based on N=2? Individual data points should be provided to assess the variability of the observations, and more biological replicates are needed, ideally using complementary techniques. Overall I feel that the presented experimental data is not very convincing and requires more backing.

Thank you for pointing out the missing information on the gene silencing experiments. The figures have been corrected/completed. The knockdown of gene expression in figure 4a is based on an n=5. We have reviewed and rewritten the legends to figures 4a to 4c to be clearer. In figure 4c we now also provide the data for the non-insulin stimulated leptin secretion in the knockdown. We now also provide a figure with the individual data points in supplementary materials (Supplementary Fig. 13).

What background gene set was used in the IPA network/gene set enrichment analysis? How was multiple testing accounted for? P-values on the order of 1E-4 are typically seen in IPA analyses and may well be false positives. Also, the plotted networks provided as supplemental data are not very helpful as there is no information on the supporting data (which can only be accessed in an active IPA session).

The background gene-set was limited to genes coding for 1,129 proteins available in our study. Multiple-testing was not taken into account in the previous version, we now adjust for multiple testing issue. Following your additional comment (see below), we expanded our analysis to all mammals (human, mouse and rats). Using Benjamini-Hochberg correction, we highlighted the coagulation pathway (adjusted p=0.049), previously reported in our initial draft. Other pathways were considered as marginally significant. By expanding our analysis to more organisms, we were able to uncover 3 large major networks. All these results were added to the revised manuscript page 6:

"Canonical pathway analysis identified the coagulation system pathway as modulated by this subset of 42 protein-coding genes ($p=2.53 \times 10^{-4}$ and corrected $p=0.049$) affected by obesity. Ingenuity Pathway Analysis (IPA) identified three major networks of interacting proteins. The first and largest network (Supplementary Fig. 1a) included 20 proteins from the list of 42 proteins and was predicted to be involved in tissue morphology, inflammatory response and cardiovascular diseases including proteins coded by genes already associated with obesity (INS, LEP, SERPINE1, IL1RAP and RARRES2). The second network (n=10 proteins) was predicted to be involved in organismal injury and abnormalities, embryonic development and organismal development, (Supplementary Fig. 1b) the third network (n=7 proteins) to developmental disorder, endocrine system disorders, hereditary disorder (Supplementary Fig. 1 c)."

The method section was also improved (page 21):

"A Benjamini-Hochberg correction for multiple testing was applied and only pathways that met the pvalue cut-off $p < 0.05$ were considered. Default parameters were used limiting source of data from mammalian studies in the Ingenuity Knowledge Base."

We agree with the reviewer that without an active IPA session some information may lack. However information from IPA networks demonstrate a high degree of connectivity between BMI-associated proteins impacted by a LCD intervention providing additional information to the reader and future researches in the field. We proposed to keep these networks as Supplementary Material.

Specific comments:

According to an update by Somalogic, the aptamer that is supposed to target C1s actually Complement Factor H: "Mass spectrometry identifies Complement Factor H as the protein CFH, enriched targets from plasma when the SOMAmer reagent 3590-8 is used for affinity capture." See: <http://www.somalogic.com/wp-content/uploads/2017/01/Revision-Announcement-of-the-1.3k-Version-SOMAscan-Assay-v2.pdf>

This update from December 2016 was unfortunately not available at the time of the analysis. We thank the reviewer for this information. This explains our observation about a pQTL SNP acting on C1s and factor H. The corresponding section describing pQTL for these 2 proteins was excluded from the paper. However we did not modify the number of available proteins and fix this number to the number of available proteins on the assay at the time of the study. This information was specified in the Method section of the manuscript (page 19):

"At the time of the analysis, 1,129 proteins were available with the Somalogic Assay. A revision was published recently by the company removing five reagents from future assay results reporting: Alkaline phosphatas, Complement C1s subcomponent, Desmoglein-2, Reticulon-4 and Tumor necrosis factor receptor superfamily member 25. Results from this proteins were kept in the current analysis but not interpreted in the manuscript."

line 242: duplicated word "LCD"

line 349: "amonacids" should read "amino acids"

Typos were corrected.

line 438: report how many participants were excluded

We now provide these numbers and identified a mistake in the sample size before/after data cleaning. The updated information is available page 18:
"Protein data were available for 530 Diogenes participants before and after the intervention. Extreme outliers were set as missing if out of protein values ± 6 IQR. Participants were excluded ($n = 18$) if more than 5% proteins were outside ± 4 SDs at baseline or after LCD. Proteins data were log2 transformed for normality for 512 informative Diogenes participants."

line 441: report how much of the variance is explained by the first/first three principal components

Estimated variances by principal components were added page 18:
"The three first components explained up to 60% of the variance, respectively 49%, 8% and 3%."

line 525: why limit IPA to "human studies" ? Most information about protein function comes from mouse knock-out studies

IPA analysis was modified to all available mammalian studies (human, mouse and rat) as described previously in this document.

line 534: report the result of performing GWAS with/without BMI as confounder

We thank the reviewer to underline this oversight. Pvalue from BMI-adjusted analyses were added to pQTL results Table 1, 2 and 4. Significant association results were not affected by addition of BMI variable to the linear mixed model used for association analysis as specified now in the Results section (page 6 and 8).

Reviewer #3 (Remarks to the Author):

The manuscript by Carayol et al reports a large-scale protein quantitative trait locus (pQTL) analysis based on a predetermined set of 1,129 plasma proteins from 494 obese subjects before and after a weight loss intervention. The study identified 55 BMI-associated cis- and trans- pQTL's at baseline and 3 trans-pQTLs after a 6 week 800 caloric diet (LCD) intervention. The study identified strong cis- or trans- pQTLs for proteins that have been previously associated with obesity, including two inflammation-related genes (IL-1 R AcP and IL-16), sE-Selectin and leptin. LCD intervention activated primarily trans-pQTLs. A novel trans-pQTL near FAM46A for leptin after LCD intervention was identified. The pQTL, however, had no detectable effect on leptin expression at baseline. In vitro study showed that knockdown of FAM46A increased insulin-stimulated leptin secretion in SGBS human adipocytes.

Thank you for your review. We provide additional details and response to all your questions point-by-point below.

Major points:

1. The timing of the acquisition of the plasma samples relative to meals should be stipulated. As should the weight status of the subjects: that is, in the LCD portion of the study were the subjects still in negative energy balance? A schematic of the clinical protocol in this regard would be helpful.

Blood sampling was made under fasting condition. This information was added to the manuscript page 18:

"A set of 1,129 proteins was measured and quantified directly from plasma sample under fasting condition before and after intervention."

By design all subjects were in negative energy balance since LCD was only 800kcal/d. A flowchart for individuals' selection from the DiOGenes cohort is now provided as Supplementary Figure 14.

2. Additional details regarding the acquisition of adipose tissue for expression analyses should be given, including weight status relative to the LCD and sites analyzed.

All details regarding adipose tissue acquisition were already provided in Viguerie et al (2012). The reference has been added to the paper in the Method section (page 19): *"Abdominal subcutaneous adipose tissue sampling were obtained by needle aspiration under local anesthesia after an overnight fast at baseline, at the end of LCD (see Viguerie et al⁷ for more information)."*

3. How does FAM46A knockdown affect leptin secretion in non-insulin stimulated SGBS cells? Because the SNP near FAM46A appeared to only affect leptin level after the LCD intervention, when insulin levels are likely to be lower, this set of data would be important to a better understanding of the role of FAM46A in regulating leptin level in LCD condition.

We have added the results for the non-insulin stimulated leptin secretion in figure 4c. The direction of the response is similar (19% reduction of measured secreted leptin in the knockdown vs 49% in the insulin stimulated condition), although it does not

reach statistical significance probably due to the low levels of leptin secretion at baseline.

4. Again, considering that the pQTL SNP next to FAM46A had no association with plasma leptin concentrations at baseline and that FAM46A expression is increased after low caloric diet intervention, the in vitro confirmation study should also examine the effect of FAM46A overexpression on basal and insulin-stimulated leptin expression and secretion in SGBS cells.

We agree that in principal the overexpression would be an interesting experiment to perform. However, a transient overexpression should be done in fully differentiated adipocytes when FAM46A expression is low and leptin expression is high. The SGBS cells (and adipocytes in general) do not respond well to transfection once fully differentiated. The solution would be a stable overexpression cell line transfected early during the differentiation process, which we feel would be out of scope for the current article.

5. What were the directions of the effect of eQTL and pQTL SNPs on FAM46A expression and plasma leptin concentrations? Were they directionally consistent with the proposed regulatory role of FAM46A on leptin production/secretion?

As proposed by the reviewer we extracted pQTL and eQTL association signal for the 2 leptin pQTL SNPs and observed expected opposite direction. We enlarged analysis to all SNPs under the pQTL association signal in the FAM46A region and add a paragraph commenting our observations in the Results section (page 11):

"To complete the genetic characterization of the QTL results we evaluated the directionality of pQTL and eQTL SNPs. Focusing on the 2 pQTL SNPs significantly associated to leptin change during LCD first. SNP rs9344031 was positively associated to leptin protein level change ($p = 1.48 \times 10^{-7}$) and negatively associated to FAM46A gene expression change ($p = 4.9 \times 10^{-3}$). The same trend was found for the second pQTL SNP rs481777 although FAM46A gene expression change did not reach a nominal significance level ($p = 0.12$) likely because of the difference in sample size between the eQTL and pQTL study. In an enlarged analysis, pQTL signals were extracted in the promoter region of FAM46A located between 81.3 and 81.8 Mb on chromosome 6 (Figure 3) for all genotyped and imputed SNPs. Of 1,833 SNPs in this region, 94 were identified as pQTLs and eQTLs for leptin and FAM46A respectively assuming an uncorrected 5% nominal significance level. All QTL SNPs displayed opposite association direction between leptin protein and FAM46A gene consistent with the regulatory role of FAM46A on leptin observed in SGBS cells.

6. Do the authors have opinions regarding why PUR8, ferritin and CA1 levels are correlated with the same pQTL?

Thank you for raising this very interesting point. All three proteins play a role in uric acid and ferritin metabolism. Both uric acid and ferritin have been shown to be markers of the progression of non-alcoholic fatty liver disease (NAFLD), a frequent co-morbidity in obese subjects. It is intriguing that the most likely gene responsible for the trans-effect i.e. TIPARP showed elevated levels in the livers of mice fed a high-fat diet under which they usually develop fatty livers.

We have added a paragraph discussing this relationship in the discussion (page 14):

" We also identified at least one trans-pQTL that influences the levels of at least three distinct proteins. A locus on chromosome 3 was significantly associated with

the levels of PUR8 encoded by the ADSL gene on chromosome 22 involved in uric acid metabolism; ferritin encoded by FTH1 on chromosome 11 involved in iron storage and CA1, a carbonic anhydrase located on chromosome 8. Comparison of results from our eQTL analysis for all genes in the region identified TIPARP as the gene with the strongest association signal. This gene was already identified in a genome-wide meta-analysis as a potential candidate for the regulation of circulating leptin levels and its expression was increased in subcutaneous adipose tissue and liver of mice fed with high-fat diet⁶. The functional relationship between these proteins is not known. However both uric acid and ferritin (the levels of which are determined in part by the associated proteins) have been shown to be markers for non-alcoholic fatty liver disease (NAFLD) and its progression, a major and frequent complication in obese individuals⁷⁵. Carbonic anhydrase (CA) is also a major regulator of uric acid levels and CA inhibitors are frequently used as diuretics and in the treatment of gout. It is intriguing that expression levels of the TIPARP gene are elevated in the livers of high fat fed mice. Thus one may speculate that the pQTL variant could be involved in the etiology of NAFLD through the regulation of the levels of these three proteins."

7. TIPARP was suggested to be the gene influencing the pQTL that affects ferritin and potentially leptin concentration through iron-ferritin metabolism. As indicated by the authors, a previous GWAS has identified TIPARP as a modulator of circulating leptin concentrations independent of BMI. Why was the TIPARP signal not detected as a significant pQTL for leptin in this study?

This region did not reach our significance levels (the best pQTL rs13080377 had $p = 7.8e-3$). It is likely that we did not have the necessary sample size to observe this baseline association. Our sample size was 500 subjects with proteomics and clinical data, while Kilpelainen et al. performed a GWA in the general population using 50,000 subjects.

8. Why was adiponectin, a protein produced in large amount by adipose tissue with plasma concentrations inversely related to adiposity not reported here, regardless of changes seen.

Adiponectin was associated to BMI at baseline ($p = 0.0048$) but not during LCD ($p = 0.36$). pQTL analysis at baseline did not highlight any significant signal after multiple testing correction (best association signal rs12338625 with $p\text{-value} = 6.39421e-06$ and corrected $p\text{-value} = 0.85$). Compared to other proteins strongly associated to BMI and significant pQTL results extracted in the current study, statistical results available from adiponectin did not meet statistical criteria to be discussed as a major hit in our manuscript.

9. A recent paper looking for genes modifying circulating leptin concentrations should be consulted for possible implication of FAM46A. And should in any event be mentioned in the discussion re strategies for finding such genes. Kilpeläinen, Tuomas O., et al. "Genome-wide meta-analysis uncovers novel loci influencing circulating leptin levels." Nature communications 7 (2016).

As also proposed by another reviewer, we compared our leptin pQTL results to those presented in Kilpelainen paper in the Discussion page 15 to 16:

"None of the two leptin associated pQTL SNPs or any SNPs in the surrounding region was found associated to leptin in Kilpelainen et al.⁶ meta-analysis. This study used

cohorts recruited from the general population and tested association between common genetic variants and circulating leptin level while we identified pQTL SNPs associated to leptin protein expression change during a weight-loss intervention in obese subjects. This may be due to a specific induction of the FAM46A gene/protein during weight loss. The reciprocal relationship between FAM46A and leptin is also apparent in the shift of gene expression during adipocyte differentiation. Thus the external stimulus and drastic change would be necessary to identify the relationship. We believe that this is a first example that an environmental cue (like a low calorie diet) that will induce large changes in gene and protein expression can lead to the identification of pQTL signals not detectable at baseline homeostasis."

Minor points:

1. In "Result" and Discussion", it is not clear how the pQTL SNP in CPN1 for Calpastatin is related to an eQTL for CWF19L1? Is there any known or inferred functional connection between calpastatin and CWF19L1?

We identified a strong pQTL signal for Calpastatin with SNPs located within CPN1 gene. Unfortunately, gene expression data was not available for CPN1 to allow an eQTL analysis for this gene. However, eQTL results were available for other genes in the region containing calpastatin pQTL signal. The strongest association between SNPs in this pQTL region and expression of surrounding genes was found for a gene located some distance away (150kb) called CWF19L1. To our knowledge, no connection exists between calpastatin and CWF19L1. This gene was highlighted based on statistical evidences in this particular sample. Nonetheless these results do not exclude CPN1 as the gene regulating calpastatin or other genes in the region for which gene expression was not available in our eQTL data.

3. What was the rationale for use of the X cell line, from individuals known to have primary

We assume that the question is about the use of the SGBS cell line. This is a well-established human pre-adipocyte cell line that has been shown to exhibit normal adipocyte features (unlike for example the mouse 3T3 cell line that lacks the expression of certain key adipocyte specific proteins) and can easily be differentiated into mature adipocytes. Unlike cell cultures from human primary pre-adipocytes from tissue biopsies these cells can be used through many passages and are easier to transfect than primary adipocytes.

4. Some of the gene symbol "ABO" looked like "AB0". Was there a typo? Disturbances in cell growth?

All typos were corrected.

** See Nature Research's author and referees' website at www.nature.com/authors for information about policies, services and author benefits

Reviewer #1 (Remarks to the Author):

The authors have addressed all my comments adequately and the paper has been improved. I have no further remarks.

Reviewer #2 (Remarks to the Author):

The author have made an effort to respond to most of this and the other reviewers' question. However, a major concern remains, which is an apparent ignorance of recent, and for this work very relevant papers that are using the Somalogic technology (see my specific comments below). This shortcoming needs to be remedied by extensively confronting findings from this paper with what has been published already (not merely citing these papers).

Specific comments:

Line 81: "To date only few large-scale pQTL studies have been reported (ref 20-23)" should be updated to include recent pGWAS, e.g. PMID 25147954 and 28240269.

Line 124: "Forty-two of the proteins were associated both with BMI at baseline ..." ... Supplemental Table 17 of <http://biorxiv.org/content/early/2017/05/05/134551> contains BMI-protein associations for the Somalogic platform based on N=3000 and should be used to replicate these associations.

Line 155: "GWAS pQTL analysis of the 192 proteins associated with BMI at baseline identified 57 pQTL's" The authors are apparently not aware of current pGWAS with Somalogic technology – most of these hits were already reported in PMID 28240269 and replicated in <http://biorxiv.org/content/early/2017/05/05/134551>.

Supplemental Tables: All supplemental tables that contain data should be provided in a machine-readable format, e.g. XLS, not PDF

Supplemental Figure 8: The bottom Figure contains only a single data point and no legend on the X-axis – something is wrong here

Reviewer #3 (Remarks to the Author):

All questions/points except #3 and #4 were addressed satisfactorily. The remaining questions/points are related to in vitro FAM46A manipulation experiments.

Specific points:

1. In response to the reviewer's request, the authors now include the results of FAM46A KD on leptin release in the basal state. These data indicate that knockdown of FAM46A did not have a statistically significant effect on leptin release in the basal state, although leptin release was marginally higher in FAM46A-KD cells. Thus, the new data do not fundamentally address the point raised in the previous cycle, i.e., that the insulin stimulated-condition in which in vitro FAM46A knockdown has shown an effect on leptin protein level is not an appropriate reflection of the fasted state in which an increased FAM46 expression was suggested to play role in suppressing leptin protein levels.

2. The authors did not conduct the in vitro overexpression study requested by the reviewer (Point #4 in the previous review) on the grounds of technical difficulty. The reviewer is aware of potential technical challenges of conducting the overexpression study, but feels that it is a necessary piece

of evidence to support the authors' conclusions. This experiment is especially crucial since the results of in vitro knockdown experiments did not fully corroborate a direct role of FAM46A in suppressing leptin protein levels in the fasted state (see Point #1 above).

3. In Fig 4a, it is not specified whether the reported FAM46A mRNA and protein levels were determined in the basal or in the insulin-stimulated condition. The modest reduction (< 33%) in FAM46A protein level after siRNA-mediated knockdown, relative to the 71% reduction of its mRNA level), suggests that FAM46A protein is relatively long-lived and that its levels may be regulated significantly by post-translational mechanisms. Six-hour treatment of cells with insulin may differentially affect FAM46A mRNA and/or protein levels in control and FAM46A knockdown cells. Thus, FAM46A protein and mRNA levels in the basal and insulin-stimulated conditions at the times when the effects on leptin level were examined should be reported.

4. There is a possibility that FAM46A, which acts in the TGF signaling pathway, may affect the differentiation status or secretory capacity of SGBS cells in general rather than (or in addition to) specifically affecting leptin secretion per se. This issue may be addressed by examining levels of adipocyte differentiation makers, such as Pparg and Cebpa, and secretion rates of other adipokines, such as adiponectin and IL-6. Depending on the outcome of these experiments, the abstract and the main text should be modified accordingly.

Specific responses to reviewers

Reviewer #2

The author have made an effort to respond to most of this and the other reviewers' question. However, a major concern remains, which is an apparent ignorance of recent, and for this work very relevant papers that are using the Somalogic technology (see my specific comments below). This shortcoming needs to be remedied by extensively confronting findings from this paper with what has been published already (not merely citing these papers).

We thank the reviewer for her/his comments and modified our manuscript accordingly as described in the following.

Specific comments:

Line 81: "To date only few large-scale pQTL studies have been reported (ref 20-23)" should be updated to include recent pGWAS, e.g. PMID 25147954 and 28240269.

The two references were added to the manuscript as requested.

Line 124: "Forty-two of the proteins were associated both with BMI at baseline ..." .. Supplemental Table 17 of <http://biorxiv.org/content/early/2017/05/05/134551> contains BMI-protein associations for the Somalogic platform based on N=3000 and should be used to replicate these associations.

At the time of submission of our manuscript, this preprint was not available (posted on bioRxiv on May 5th 2017). Nevertheless the work by Sun and colleagues is interesting and reports in Supplementary a list of proteins associated with BMI in the general population. This contrast with our approach that investigated proteins associated with both baseline BMI and weight loss upon LCD in a stratified population.

We included a comparison of our findings (see results section p6) with the reported proteins by Sun et al. Notably, we confirm the association between our 42 identified proteins and baseline BMI (with Bonferroni adjusted p-value < 5%). Information for these proteins was extracted from the Sun et al. corresponding tables, has been added in Supplementary Table 3 and is presented in the corresponding results section (p6):

"Recently, a proteome-wide association study tested the association between BMI and 3,620 plasma proteins assayed in 3,301 healthy individuals from the general population⁴¹. The same SOMAscan technology was used but with a different and larger SOMAscan assay. In this external study, summary statistics (association with BMI) were available for 38 / 42 of our BMI-associated proteins (Supplementary Table 3). All of our 38 proteins demonstrated significant association with BMI (with Bonferroni corrected pvalue < 0.05). Association with BMI was in the same direction for all but two proteins, angiotensin-2 and GDF-11 with weakest replication signal compared to other tested proteins."

These new results are also discussed in the Discussion section (p18):

“Of forty-two proteins identified as associated both to BMI level at baseline and fold change during LCD, 38 were already investigated in a population-based study and identified as correlated to BMI⁴¹. Comparison of pQTL signals with two large independent studies^{25,41}, conducted in the general population, supported further our pQTL findings and highlighted 22 pQTL signals that were common between all three studies. Both our functional assays and comparison of association signals with studies from slightly different design, lend support to our results and stress the importance of our identified pQTL signals and their potential role in regulation of proteins like *FAM46A* and gustin.”

Line 155: “GWAS pQTL analysis of the 192 proteins associated with BMI at baseline identified 57 pQTL’s” The authors are apparently not aware of current pGWAS with Somalogic technology – most of these hits were already reported in PMID 28240269 and replicated in <http://biorxiv.org/content/early/2017/05/05/134551>.

Indeed, these two papers were not available at the time of submission. We added a comparison of these findings in our revised manuscript (see page 8-9 and Supplementary Tables 4 and 5):

“Two studies recently published performed genome-wide association test of autosomal variants against levels of Somalogic plasma proteins in population-based cohorts^{25,41}. The first study by Suhre et al.²⁵, analysed two cohorts: the population-based KORA (Cooperative Health Research in the Region of Augsburg) for discovery and the Qatar Metabolomics Study on Diabetes (QMDiab) for replication of their association signals. Eight of our pQTL SNPs were significant in KORA and four of them successfully replicated in QMDiab. We also compared our results with the second study by Sun et al.⁴¹. This highlighted two additional pQTLs (Supplementary Table 4).

Since these comparisons were limited to the presence of the same SNP marker, we extended it to a locus-based comparison (i.e. QTL signals). This was made by extending association signal to pQTL SNPs and SNPs in moderate LD ($r^2 > 0.2$ based on 1,000 Genomes Project genotype European reference population). This approach showed that 39 and 34 QTL signals were in common with those reported by Suhre et al.²⁵ and Sun et al.⁴¹. Specifically, 22 QTL signals (20 cis- and 2 trans-acting) were common between our study and the two external studies (Supplementary Table 5). Among this list of common QTL signals, we found IL-1 R AcP previously described and carbonic anhydrase 6, also called gustin, largely discussed later in this paper.”

Page 15 of the Discussion clarifies the differences between our study and the two independent studies:

“To our knowledge, this is the first pQTL analysis in obese subjects, following a restricted caloric intervention. Previous studies focused on association between baseline protein levels and BMI in the general population. “

Supplemental Tables: All supplemental tables that contain data should be provided in a machine-readable format, e.g. XLS, not PDF

Our supplementary tables were provided as XLS but were automatically converted as PDF by the submission system. We will notify the editor and enquire whether Supplementary Tables can be provided and published as XLS.

Supplemental Figure 8: The bottom Figure contains only a single data point and no legend on the X-axis – something is wrong here

We agree with the reviewer that these 2 figures lack of clarity and may confuse the readers. Supplementary Figures 8 and 9 were drastically simplified. Previous versions presented results for all SNPs (top panels) and only SNPs with significant p-value after correction for multiple-testing (bottom panel). The new figures only show unadjusted association results, in the form of Manhattan plots for gustin (Supplementary Figure 8) and chemerin (Supplementary Figure 9). The adjusted p-values are provided in the Results section (p10), with the reference to the corresponding supplementary figure.

Reviewer #3

1. In response to the reviewer's request, the authors now include the results of FAM46A KD on leptin release in the basal state. These data indicate that knockdown of FAM46A did not have a statistically significant effect on leptin release in the basal state, although leptin release was marginally higher in FAM46A-KD cells. Thus, the new data do not fundamentally address the point raised in the previous cycle, i.e., that the insulin stimulated-condition in which in vitro FAM46A knockdown has shown an effect on leptin protein level is not an appropriate reflection of the fasted state in which an increased FAM46 expression was suggested to play role in suppressing leptin protein levels.

We thank the reviewer for his comment on the basal vs insulin stimulated conditions. As the reviewer states the effect of the *FAM46A* knockdown is somewhat weaker in the basal state than in the insulin stimulated state. The reviewer concludes that therefore the cell experiments are not in line with the findings from the pQTL study that show that there is a reverse relationship between *FAM46A* levels (controlled by a genetic variant) and leptin levels. We respectfully disagree with the reviewer on this point. The cell experiments were designed to investigate whether *FAM46A* directly influences leptin secretion and if this observation was specific (and we are thankful for the reviewer's suggestions to test other secreted proteins and differentiation markers in this context). Basal leptin secretion from adipocytes is low and insulin has been shown to be a necessary stimulus for leptin secretion (see for example Cammisoto & Bukowieki, Am J Physiol Cell Physiol. 2002 Jul;283(1):C244-50.). In no case can these cell experiments be seen as proxies to simulate an 8-week low caloric intervention. Although insulin levels will generally be lower after the LCD both at fasting and post-prandial, this is accompanied by an increased insulin sensitivity thus the amplitude of insulin stimulated responses is indeed greater after the LCD intervention than before. We therefore believe that our experiments do address the question if *FAM46A* has direct effects on leptin secretion irrespective of the fact that only the insulin stimulus shows nominally

significant results (although from a pure statistical point of view the marginal p-value for the basal state suggests that there is also an effect on leptin).

2. The authors did not conduct the *in vitro* overexpression study requested by the reviewer (Point #4 in the previous review) on the grounds of technical difficulty. The reviewer is aware of potential technical challenges of conducting the overexpression study, but feels that it is a necessary piece of evidence to support the authors' conclusions. This experiment is especially crucial since the results of *in vitro* knockdown experiments did not fully corroborate a direct role of FAM46A in suppressing leptin protein levels in the fasted state (see Point #1 above).

We thank the reviewer for restating the importance of the overexpression experiments and to acknowledge the technical challenges. We have now added the overexpression experiment. The results (see figure below) show that overexpression leads indeed to a significant decrease of leptin secretion by the adipocytes both at the basal as well as the insulin stimulated state. We have added the respective sections in the results and methods parts of the paper (p13):

“To overexpress *FAM46A* SGBS cells were transfected with 0.5µg of full length *FAM46A* cDNA. This resulted in a 21-fold increase of *FAM46A* expression (Supplementary Fig. 16). *FAM46A* overexpression resulted in a significant 28% and 24% (p= 0.034 and 0.038, one-sided t test, for both basal and insulin stimulated states respectively) reduction of insulin and non-insulin stimulated leptin release respectively (Fig. 5B).”

Figures are provided in the manuscript and Supplementary Material as follow:

Supplementary figure 16

Overexpression of *FAM46A* in SGBS adipocytes. SGBS cells were transfected with 0.5 µg full length *FAM46A* cDNA for 48h and resulted in a 20-fold increase of *FAM46A* expression. *FAM46A* mRNA was assessed by qRT/PCR at day nine. Gene expression levels are expressed relative to *TBP* (TATA Binding Protein) expression. The data are shown as the means +/- SD (n=3) with *P<0.05, **P<0.01 for statistical significance calculated using the Student's t test ($p_{\text{Basal}} = 3.0 \times 10^{-4}$, $p_{\text{Insulin}} = 1.0 \times 10^{-4}$).

Figure 5. Effects of *FAM46A* alterations (knockdown and overexpression) on leptin secretion. (A) leptin release in the *FAM46A* knockdown. (B) leptin release upon overexpression of *FAM46A*. Nine-day-old differentiating preadipocytes were cultured in serum-free basal medium without insulin or after stimulation with 100 nM insulin for 6 hours. Conditioned medium was removed and the secreted leptin content was measured using a DuoSet ELISA kit. Accumulated leptin was normalized to total amount of protein from each well. Results are given as the mean \pm SD (n=11) with *P<0.05, **P<0.01 for statistical significance calculated using a unilateral Student's *t* test for an increased or a decreased leptin release according to a knockdown or an overexpression of *FAM46A*, respectively.

B

3. In Fig 4a, it is not specified whether the reported FAM46A mRNA and protein levels were determined in the basal or in the insulin-stimulated condition. The modest reduction (< 33%) in FAM46A protein level after siRNA-mediated knockdown, relative to the 71% reduction of its mRNA level), suggests that FAM46A protein is relatively long-lived and that its levels may be regulated significantly by post-translational mechanisms. Six-hour treatment of cells with insulin may differentially affect FAM46A mRNA and/or protein levels in control and FAM46A knockdown cells. Thus, FAM46A protein and mRNA levels in the basal and insulin-stimulated conditions at the times when the effects on leptin level were examined should be reported.

As requested by the reviewer we have amended figure 4A and present both the basal and insulin stimulated values for *FAM46A* expression for the wildtype and knockdown. We have also added a figure with all individual time-points as requested, both for *FAM46A* and leptin gene expression as a new supplementary figure 13A & B. We show that leptin gene expression is indeed not altered at any condition. We also confirm the *FAM46A* gene expression and protein levels were determined at the same time-points as the measurements of leptin. We have added an appropriate clarification in the methods “*FAM46A* overexpression” chapter page 25 and 26 (Section highlighted in blue).

Figures are provided in the manuscript and Supplementary Material as follow:

Figure 4

***FAM46A* gene silencing effect on leptin protein expression in SGBS**

adipocytes. SGBS cells were transfected with negative control siRNA (siNEG) or *FAM46A*-specific siRNA (si*FAM46A*) and experiments were performed 48 h after transfection. Gene expression and protein measurements for *FAM46A* were conducted at the basal and insulin stimulated states at the same time as leptin expression/secretion analyses at day nine. (A) Gene silencing of *FAM46A* in SGBS adipocytes. Gene expression levels are expressed relative to TBP (TATA Binding Protein) gene expression for *FAM46A*. Results are given as the mean +/-SD for basal or insulin stimulated state (n=9); (B) Western blot for *FAM46A* protein expression in SGBS adipocytes in wildtype and *FAM46A* knockdown. *FAM46A* expression was detected in cell lysate by Western blotting. Protein expression levels are expressed relative to Actin expression (n=2). (C) Effects of *FAM46A* knockdown on leptin gene expression at basal non-insulin stimulated conditions and after stimulation with 100 mM insulin. Gene expression levels are expressed relative to TBP gene expression. Results are given as the mean +/-SD (n=9). *P<0.05, **P<0.01 are provided for statistical significance calculated using a Student’s t test.

A

Supplementary Fig. 13

FAM46A gene silencing effect on leptin protein expression in SGBS adipocytes (individual datapoints corresponding to fig. 4A and 4B). SGBS cells were transfected with negative control siRNA (siNEG) or FAM46A-specific siRNA (siFAM46A) and experiments were performed at day nine. Gene expression levels are expressed relative to TBP (TATA Binding Protein) expression. Results are given as the mean \pm SD for n=9.

4. There is a possibility that FAM46A, which acts in the TGF signaling pathway, may affect the differentiation status or secretory capacity of SGBS cells in general rather than (or in addition to) specifically affecting leptin secretion per se. This issue may be addressed by examining levels of adipocyte differentiation makers, such as Pparg and Cebpa, and secretion rates of other adipokines, such as adiponectin and IL-6. Depending on the outcome of these experiments, the abstract and the main text should be modified accordingly.

This is a very important remark. We now provide data for PPAR γ and CEBP α expression both in the knockdown as well as the overexpression model and

show that expression of the differentiation markers is not different between wildtype and models. We also show that secretion of adiponectin is not altered by either knockdown or overexpression. As suggested by the reviewer we also attempted to measure IL6 secretion. However, as can be seen from supplementary figure 12 IL6 was found to be very low in SGBS and below the detection limit of the ELISA method (sensitivity IL6= 0.01 pg/ml). Appropriate sections have been added to results (p12 and 13):

“In order to evaluate if *FAM46A* levels directly influenced leptin levels, we performed knockdown and overexpression studies in the SGBS human adipocyte cell line. For the knockdown experiments a specific siRNA for the *FAM46A* gene was designed. The transfection with the siRNA resulted in a 70% and 74% reduction of *FAM46A* gene expression in the basal and insulin stimulated state respectively. *FAM46A* protein was reduced by 31% and 73% in the basal and insulin induces state respectively (Fig. 4A, Fig. 4B and Supplementary Fig. 13). *FAM46A* knockdown had no influence on leptin gene expression (Fig. 4C) and did not show any effect on markers of adipocyte differentiation PPAR γ and CEBP α (Supplementary Fig. 14). Conversely, *FAM46A* knockdown resulted in a highly significant 49% increase of insulin stimulated leptin secretion from the adipocytes (p=0.0002 one-sided *t* test for leptin increase). Under conditions without prior insulin stimulation the knockdown still showed a marginal 19% increase in leptin secretion (p=0.065 one-sided test for leptin increase; Fig. 5A). To evaluate if the effects of the knockdown of *FAM46A* were specific for leptin we tested the secretion of adiponectin. Secretion of adiponectin was not affected by the knockdown (Supplementary Fig. 15). To overexpress *FAM46A* SGBS cells were transfected with 0.5 μ g of full length *FAM46A* cDNA. This resulted in a 21-fold increase of *FAM46A* expression (Supplementary Fig. 16). *FAM46A* overexpression resulted in a significant 28% and 24% (p= 0.034 and 0.038, one-sided *t* test, for both basal and insulin stimulated states respectively) reduction of insulin and non-insulin stimulated leptin release respectively (Fig. 5B). Overexpression did not influence PPAR γ or CEBP α gene expression (Supplementary Fig. 17). Adiponectin release was not affected by *FAM46A* overexpression both in the basal as well as the insulin stimulated state (Supplementary Fig. 18).”

and to discussion (p18):

“Conversely overexpression of *FAM46A* resulted in a significant reduction of leptin secretion in line with our hypothesis. Both knockdown as well as overexpression did not alter PPAR γ and CEBP α levels or adiponectin secretion, indicating that the changes seen in leptin secretion are not due to effects of *FAM46A* on adipocyte differentiation and that the effects seem to be specific for leptin. “

All figures are provided in the manuscript and Supplementary Material as follow:

Figure 4

***FAM46A* gene silencing effect on leptin protein expression in SGBS**

adipocytes. SGBS cells were transfected with negative control siRNA (siNEG) or

FAM46A-specific siRNA (siFAM46A) and experiments were performed 48 h after transfection. Gene expression and protein measurements for *FAM46A* were conducted at the basal and insulin stimulated states at the same time as leptin expression/secretion analyses at day nine. (A) Gene silencing of *FAM46A* in SGBS adipocytes. Gene expression levels are expressed relative to TBP (TATA Binding Protein) gene expression for *FAM46A*. Results are given as the mean \pm SD for basal or insulin stimulated state (n=9); (B) Western blot for *FAM46A* protein expression in SGBS adipocytes in wildtype and *FAM46A* knockdown. *FAM46A* expression was detected in cell lysate by Western blotting. Protein expression levels are expressed relative to Actin expression (n=2). (C) Effects of *FAM46A* knockdown on leptin gene expression at basal non-insulin stimulated conditions and after stimulation with 100 mM insulin. Gene expression levels are expressed relative to TBP gene expression. Results are given as the mean \pm SD (n=9). *P<0.05, **P<0.01 are provided for statistical significance calculated using a Student's t test.

A

B

C

Figure 5. Effects of *FAM46A* alterations (knockdown and overexpression) on leptin secretion. (A) leptin release in the *FAM46A* knockdown. (B) leptin release upon overexpression of *FAM46A*. Nine-day-old differentiating preadipocytes were cultured in serum-free basal medium without insulin or after stimulation with 100 nM insulin for 6 hours. Conditioned medium was removed and the secreted leptin content was measured using a DuoSet ELISA kit. Accumulated leptin was normalized to total amount of protein from each well. Results are given as the mean \pm SD

(n=11) with *P<0.05, **P<0.01 for statistical significance calculated using a unilateral Student's *t* test for an increased or a decreased leptin release according to a knockdown or an overexpression of *FAM46A*, respectively.

A

B

Supplementary Fig. 13

***FAM46A* gene silencing effect on leptin protein expression in SGBS adipocytes**

(individual datapoints corresponding to fig. 4A and 4B). SGBS cells were transfected with negative control siRNA (siNEG) or *FAM46A*-specific siRNA (siFAM46A) and experiments were performed at day nine. Gene expression levels are expressed relative to TBP (TATA Binding Protein) expression. Results are given as the mean +/-SD for n=9.

Supplementary figure 14

Expression levels of PPAR γ and CEBP α in the wildtype and knockdown SGBS adipocytes. SGBS cells transfected with negative control siRNA (siNEG) or FAM46A-specific siRNA (siFAM46A). PPAR γ and CEBP α mRNA was assessed by quantitative real-time PCR (qRT-PCR) at day nine. Gene expression levels are expressed relative to TBP (TATA Binding Protein) gene expression. The data are shown as the means \pm SD (n=5). *FAM46A* knockdown did not affect either PPAR γ or CEBP α expression levels (Student's *t* test $p_{\text{ppar}\gamma}$ = 0.4789, $p_{\text{CEBP}\alpha}$ = 0.0821, n=6).

Supplementary figure 15

Secretion of adiponectin in the wildtype and knockdown SGBS adipocytes. SGBS cells were transfected with negative control siRNA (siNEG) or FAM46A-specific siRNA (siFAM46A). Adiponectin secretion was determined by enzyme-linked immuneabsorbant assay (ELISA) at baseline or after stimulation with 100 nM insulin for 6 hours. The data are shown as the means \pm SD (n=3) for statistical significance calculated using the Student's *t* test. Adiponectin secretion did not differ between wildtype and *FAM46A* knockdown (Student's *t* test p_{Basal} = 0.3453, p_{Insulin} = 0.1095, n=3).

Supplementary figure 16

Overexpression of *FAM46A* in SGBS adipocytes. SGBS cells were transfected with 0.5 μ g full length *FAM46A* cDNA for 48h and resulted in a 20-fold increase of *FAM46A* expression. *FAM46A* mRNA was assessed by qRT/PCR at day nine. Gene expression levels are expressed relative to *TBP* (TATA Binding Protein) expression. The data are shown as the means \pm SD (n=3) with *P<0.05, **P<0.01 for statistical significance calculated using the Student's *t* test ($p_{\text{Basal}} = 3.0 \times 10^{-4}$, $p_{\text{Insulin}} = 1.0 \times 10^{-4}$).

Supplementary figure 17

Expression levels of PPAR γ and CEBP α in the wildtype and *FAM46A* overexpressing SGBS adipocytes. mRNA levels of PPAR γ and CEBP α were measured by qRT-PCR. Gene expression levels are expressed relative to *TBP* (TATA Binding Protein) expression. PPAR γ and CEBP α levels did not differ between wildtype and *FAM46A* overexpressing cells ($p_{\text{PPAR}\gamma} = 0.9813$, $p_{\text{CEBP}\alpha} = 0.3939$, n=5). Data are shown as the means \pm SD.

Supplementary figure 18

Secretion of adiponectin in the wildtype and *FAM46A* overexpressing SGBS

adipocytes. SGBS cells were transfected with 0.5 μ g of empty vector (Control) or *FAM46A*-specific cDNA (*FAM46A*). Adiponectin and IL6 secretion were determined by enzyme-linked immunoabsorbant assay (ELISA) at baseline or after stimulation with 100 nM insulin for 6 hours. Adiponectin secretion did not differ between wildtype and *FAM46A* knockdown (Student's *t* test $p_{\text{Basal}} = 0.26$, $p_{\text{Insulin}} = 0.11$, $n=3$). The data are shown as the means \pm SD.

REVIEWERS' COMMENTS:

Reviewer #2 (Remarks to the Author):

The authors answered all remaining questions.

Reviewer #3 (Remarks to the Author):

The authors have conducted additional experiments that this reviewer has suggested, including overexpression of FAM46A in SGBS adipocytes and assessment of effects of the manipulations of FAM46 expression on adipocyte differential status and secretory function as controls. The new data support the conclusion of the study. The revision is responsive.